# GeLoRA: Geometric Adaptive Ranks For Efficient LoRA Fine-tuning

## Abstract

Fine-tuning large language models (LLMs) is computationally intensive because it requires updating all parameters. Low-Rank Adaptation (LoRA) improves efficiency by modifying only a subset of weights but introduces a trade-off between expressivity and computational cost: lower ranks reduce resources but limit expressiveness, while higher ranks enhance expressivity at increased cost. Despite recent advances in adaptive LoRA techniques, existing methods fail to provide a theoretical basis for optimizing the trade-off between model performance and efficiency. We propose Geometric Low-Rank Adaptation (GeLoRA), a novel framework that computes the intrinsic dimensionality of hidden state representations to adaptively select LoRA ranks. We demonstrate that the intrinsic dimension provides a lower bound for the optimal rank of LoRA matrices, allowing for a principled selection that balances efficiency and expressivity. GeLoRA dynamically adjusts the rank for each layer based on the intrinsic dimensionality of its input and output representations, recognizing that not all model parameters equally impact fine-tuning. Empirical validation on multiple tasks shows that GeLoRA consistently outperforms recent baselines within the same parameter budget.

## 1 Introduction

LLMs are currently at the forefront of natural language processing tasks, yet achieving effective personalization requires additional fine-tuning. Pretraining an LLM on a diverse corpus enables it to learn general linguistic patterns and representations, which can be further refined through fine-tuning on task-specific datasets. However, fine-tuning the entire model is computationally expensive, both in terms of time and memory. To address this, a more efficient approach involves adjusting only a subset of the model's parameters, known as Parameter-Efficient Fine-Tuning (PEFT) (Han et al., 2024). PEFT methods include techniques such as adapter layers (Houlsby et al., 2019), which introduce new trainable layers into the model's backbone, and approaches like BitFit (Zaken et al., 2022), which modify a subset of the model's original weights (e.g. bias weights). Low-rank adaptation methods, such as LoRA (Hu et al., 2021), decompose update matrices into low-rank components and are particularly prominent in reducing computational costs, while maintaining comparable performance to full fine-tuning.

LoRA and its variants operate under the assumption that pre-trained language models possess a low "intrinsic dimension" (Aghajanyan et al., 2020; Li et al., 2018), suggesting that weight updates should similarly exhibit low rank. However, a key challenge with these techniques lies in determining the optimal rank values, which involves balancing expressivity and computational efficiency. Expressivity refers to the model's ability to capture complex patterns in the data, while computational efficiency pertains to the speed and resource requirements for fine-tuning. The trade-off is evident: lower ranks reduce expressivity but enhance memory efficiency and computational speed, whereas higher ranks increase expressivity at the cost of greater memory usage, longer computation times, and most likely more data to learn weights reliably. Typically, ranks are set uniformly across all layers, with practitioners relying on trial-and-error to achieve a balance between expressivity and efficiency. This process is time-consuming and may not always yield optimal results.

On the other hand, using random projection to reduce the dimensionality of the parameter space until achieving 90% of the full fine-tuning performance may not be ideal, as it inherently limits the model's potential to achieve higher performance. Recent studies on the geometry of hidden

representations (Valeriani et al., 2023) reveal that these representations also exhibit low intrinsic dimensionality, reflecting the compression occurring at each layer of the model. This raises a natural question:

> *Is there a connection between the manifold of data representations and the manifold of model parameters?*

We theoretically investigate the relationship between the intrinsic dimensionality of data representations and the ranks of weight updates in language models, deriving a lower bound for the optimal rank based on the intrinsic dimensionalities of the input and output of each transformer block. Building on this foundation, we propose a novel approach, ***Geometric Low-Rank Adaptation (GeLoRA)***, to address the trade-off between expressivity and computational efficiency by exploiting the geometric properties of the model's hidden representations. GeLoRA leverages intrinsic dimensionalities to provide a more principled mechanism for adjusting ranks, thereby achieving an optimal balance between model expressivity and computational constraints. Our method dynamically adjusts the ranks for low-rank adaptation by considering both the compression occurring at each transformer block and the specific characteristics of the model and dataset, offering a more precise and theoretically motivated balance between performance and resource efficiency.

Determining the ground truth intrinsic dimension of each hidden state is impractical; however, various techniques can provide reliable estimates. Among these, we will adopt the Two Nearest Neighbors (TwoNN) method (Facco et al., 2017), which has proven to be an effective estimator. It is robust to variations in curvature and density within the data and has been widely used to analyze representations in deep neural networks in previous studies (Ansuini et al., 2019; Doimo et al., 2020; Valeriani et al., 2023; Cheng et al., 2023; Kvinge et al., 2023; Basile et al., 2024).

**Contributions.** The contributions of our work are as follows:

- **Theoretical Framework for LoRA Effectiveness:** We establish a theoretical framework that explains the effectiveness of LoRA. Specifically, we derive a theoretical lower bound that connects the intrinsic dimensionalities of the data representation manifolds at the inputs and outputs of transformer blocks with the ranks of their constituent layers.

- **Introduction of the GeLoRA Approach:** Building upon the derived lower bound, we introduce the GeLoRA approach, which dynamically adjusts the LoRA ranks across model weights to better align with the intrinsic dimensionalities of data representations.

- **Empirical Validation of GeLoRA:** Through extensive experiments and analyses, we validate the practical performance and efficiency of the GeLoRA framework. Our results demonstrate that GeLoRA outperforms existing baselines while maintaining the same parameter budget.

## 2 RELATED WORK

LLMs have achieved state-of-the-art performance in a wide range of natural language processing (NLP) tasks across diverse domains. Models such as GPT (Brown et al., 2020) and BERT (Devlin et al., 2019) have demonstrated exceptional proficiency in tasks including language modeling, sentiment analysis, machine translation, and question answering, which showcases their versatility in natural language understanding and generation.

However, developing a more personalized model requires additional fine-tuning, which must be handled efficiently due to the substantial computational costs involved. This is where PEFT (Han et al., 2024) comes into play. It aims to balance the fine-tuning performance with the need to reduce computational overhead by selectively adjusting a small subset of the model's parameters, thereby minimizing resource consumption, as compared to the more resource-intensive process of full fine-tuning.

Within this framework, different lines of research in model fine-tuning explore various approaches to optimizing efficiency. One such approach focuses on parameter tuning techniques, where only a subset of model parameters is trained while others remain fixed. An example is BitFit (Zaken et al., 2022), which exclusively adjusts the bias terms and the task-specific head within the model, leaving the remaining parameters unchanged. Another research direction involves the use of adapter layers

by introducing small trainable layers, known as "adapters" (Houlsby et al., 2019), into the model, which enable adaptation to new tasks without altering the model's original weights. Moreover, context-based fine-tuning methods (Petrov et al., 2024) are used to influence model outputs through input representation modification. Prefix tuning (Li & Liang, 2021), for instance, appends task-specific parameters to the input's embedding, guiding the model's responses without altering its core parameters. Finally, LoRA (Hu et al., 2021; Dettmers et al., 2023; Hayou et al., 2024) represents a significant line of research that involves decomposing update matrices into the product of two low-rank matrices to reduce the number of trainable parameters, while maintaining comparable performance to full fine-tuning. Despite its advantages, LoRA faces challenges in determining the appropriate rank for the low-rank matrices. Typically, the rank is set uniformly across layers through a trial-and-error process, which is often suboptimal.

More recently, several LoRA variants have been developed to address the issue of setting uniform rank values by dynamically adjusting the rank for each layer. These variants compute importance scores or prune unnecessary ranks based on budget constraints, thereby optimizing rank alloca-tion. Notable examples include AdaLoRA (Zhang et al., 2023), SaLoRA (Hu et al., 2023), SoRA (Ding et al., 2023), and ALoRA (Liu et al., 2024), each offering strategies to improve fine-tuning efficiency. AdaLoRA dynamically allocates the parameter budget across weight matrices during fine-tuning using singular value decomposition (SVD). It adjusts the rank of matrices by assign-ing higher ranks to critical singular values and pruning less important ones, resulting in a sparse selection of ranks. However, its heuristic criterion for sparsity selection lacks strong theoretical jus-tification. Additionally, the computational complexity is increased due to operations like computing moving averages for importance scores and handling gradients from orthogonality regularization during training. On the other hand, SaLoRA dynamically learns the intrinsic rank of each incremen-tal matrix using a binary gating mechanism and a differentiable relaxation method, which selectively removes non-critical components. While this improves efficiency, removing these components may introduce instability during training. To mitigate this, orthogonality regularization is applied to the factor matrices, improving training stability and generalization. However, the optimization process, which involves Lagrangian relaxation and orthogonal regularization, increases the computational overhead. SoRA also adjusts the intrinsic rank dynamically during training by employing a sparse gating unit, which is learned through the minimization of the $l_0$ norm via the proximal gradient method. Despite its promise, the sparsifying process lacks a strong theoretical foundation and may struggle to generalize to new domains effectively. Lastly, ALoRA enables dynamic rank adjustment during the adaptation process through two key steps: first, estimating the importance scores of each LoRA rank, and then pruning less important or negatively impactful ranks while reallocating re-sources to critical transformer modules that require higher ranks. However, the computational cost of performing adaptive budget LoRA (AB-LoRA) can be high, which may hinder its practicality in certain settings.

## 3 GeLoRA: Geometric Low Rank Adaptation

### 3.1 Intuition

Consider a linear map $f : x \mapsto Wx$, where the matrix $W$ has low rank $r$. The low rank of $W$ implies that $f$ compresses the semantic information of $x$ into a lower-dimensional space, such that $\dim \Im m f = r$. While the functions approximated by transformer blocks are far more complex than a linear map, we will later show that intrinsic dimension profiles can provide valuable insight for selecting appropriate ranks for each layer of a language model. Specifically, they offer a lower bound on the number of parameters required to effectively encode information. To rigorously examine how the rank of hidden states correlates with the number of parameters needed for effective fine-tuning in a transformer block, we present a formal theoretical framework in the next section.

### 3.2 Theoretical Formulation

For clarity and consistency, we maintain the notation used in the original low-rank adaptation paper (Hu et al., 2021). Without loss of generality, we will focus on the language modeling problem, where the goal is to maximize conditional probabilities given a task-specific prompt. Each downstream task can be represented by a dataset comprising context-target pairs $\mathcal{Z} = \{(x_i, y_i)\}$, where both $x_i$ and $y_i$ are sequences of tokens. The primary objective is to accurately predict $y_i$ given $x_i$. For example,

in a summarization task, $x_i$ represents the original content and $y_i$ its summary. Mathematically, this can be modeled as follows:

$$\max_{\phi \in \Phi} \sum_{(x,y) \in \mathcal{Z}} \sum_{t=1}^{|y|} \log(\mathbb{P}_\phi(y_t \mid x, y_{<t}))$$

Here, $\Phi$ denotes the parameter set of the model, and $\mathbb{P}_\Phi(\cdot \mid \cdot)$ represents the conditional probability describing the relationship between context and target pairs. This probability distribution can be understood as a point on a neuromanifold $\mathcal{M} = \{\mathrm{NN}_\phi \mid \phi \in \Phi\}$.

The geometry of this manifold is characterized by the Fisher Information Matrix (FIM) (Fisher, 1922) with respect to $\phi$, which is given by:

$$\mathcal{I}(\phi) = \mathbb{E}_{x \sim \mathbb{P}_{\text{data}}, y \sim \mathbb{P}(\cdot \mid x; \phi)} \left[ \left( \frac{\partial}{\partial \phi} \log \mathbb{P}(y \mid x; \phi) \right) \left( \frac{\partial}{\partial \phi} \log \mathbb{P}(y \mid x; \phi) \right)^T \right]$$

The FIM defines a Riemannian metric on the learning parameter space (Amari, 2021), characterizing its curvature (Čencov, 1982). However, learning models often exhibit singularities (Watanabe, 2009), meaning that the rank of the matrix is less than its full dimension.

Transformer models typically have an extremely large number of parameters, often ranging in the millions or even billions, due to their deep and wide architectures. This high-dimensional parameter space can lead to parameter redundancy and strong correlations between parameters, as noted by Dalvi et al. (2020). Such redundancy, or multicollinearity, can result in linear dependencies among the gradients of the log-likelihood with respect to different parameters. Another motivation stems from the behavior of optimizers such as Stochastic Gradient Descent (SGD) (Ruder, 2017). These optimizers tend to prefer flatter minima during gradient descent (Jastrzębski et al., 2018), often resulting in plateaus in the gradient learning process. As a result, the FIM may exhibit eigenvalues close to zero, indicating singular or near-singular behavior.

In this context, the rank of $\mathcal{I}(\phi)$, defined by the number of non-zero eigenvalues of the FIM, reflects the number of degrees of freedom (directions) at a point $\phi$ that can modify the probabilistic model $\mathbb{P}_\Phi(\cdot \mid \cdot)$. This is often referred to as the local dimensionality (Sun & Nielsen, 2024). Figure 1 illustrates this concept, where the local dimensionality is 1, while the dimension of the space is 2.

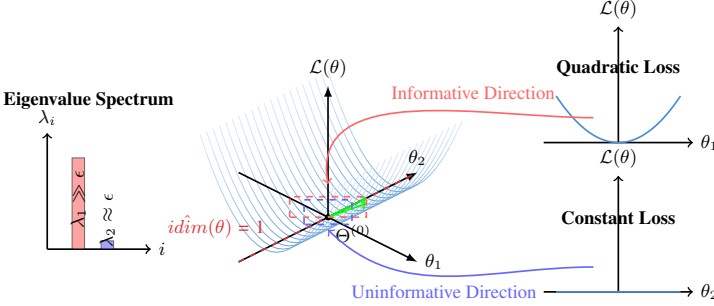

Figure 1: Assume that locally around $\Theta^{(0)}$, the loss function can be approximated by $\mathcal{L}(\theta_1, \theta_2) = \frac{1}{2}\theta_1^2$. In this scenario, the loss landscape exhibits a single free direction. The loss depends exclusively on $\theta_1$, while $\theta_2$ has no influence on it. As a result, changing $\theta_2$ alone does not affect the loss, making $\theta_2$ a free direction in the landscape. In contrast, variations in $\theta_1$ lead to changes in the loss, meaning that the zero-loss set forms a line along the $\theta_2$-axis. Therefore, the local dimensionality of the low-loss region is 1.

**Definition 3.1 (Local Dimensionality).** The local dimensionality, denoted as $d(\phi)$, is defined as the rank of the information matrix $\mathcal{I}(\phi)$. It represents the number of parameters that need to be optimized in the model, indicating the effective dimensionality of the parameter space around the point $\phi$.

Ideally, we aim to compute the local dimensionality of the parameter space at each gradient step. However, two primary challenges hinder this approach. Firstly, the information matrix behaves as a random matrix, typically maintaining full rank with probability 1 (Feng & Zhang, 2007). Secondly, the computational feasibility poses a significant obstacle, as computing the FIM at each step requires extensive computational resources.

While the FIM is almost surely of full rank, it often has very small eigenvalues, on the order of $\epsilon \in \mathbb{R}^+$. According to the Cramér-Rao bound, the variance of the parameter estimates is greater than or equal to $1/\epsilon$. Therefore, parameters associated with such small eigenvalues provide negligible information about the model and can be considered effectively uninformative. Disregarding parameters with very small eigenvalues leads us to the concept of intrinsic dimension. The intrinsic dimension is defined as the minimum number of parameters required to capture the local variance of the data points effectively. Consequently, the intrinsic dimension represents a lower bound on the local dimensionality.

**Theorem 3.1** (**Intrinsic Dimension as a Lower Bound**). *The intrinsic dimension $\hat{idim}(\phi)$ is a lower bound to the local dimensionality $d(\phi)$.*

$$d(\phi) \geq \hat{idim}(\phi).$$

Several significant challenges persist. First, the computation of the FIM and the determination of its rank are prohibitively expensive in terms of computational resources. Second, estimating the intrinsic dimension of the neuromanifold is also infeasible. Furthermore, the required number of parameters to optimize (i.e. the rank of the FIM) pertains to the entire model rather than to each independent matrix, resulting in a high lack of granularity.

However, we have access to the input data and its representations across different transformer blocks within the large language model. Consequently, we can shift our focus to the data manifold, which is subjected to a series of transformations that map it to new representations, resulting in manifolds with differing geometries. To analyze the changes in geometry, particularly the alterations in dimensionality, we will begin by defining the components of the transformer blocks. Each transformer block comprises two primary components: a multi-head attention mechanism and a feed-forward network. Additionally, it incorporates skip connections, which are essential for mitigating the rank collapse problem, and a normalization layer.

**Theorem 3.2** (**Rank Bound of Transformer Blocks**). *Let $\mathcal{M}$ denote a language model consisting of $N$ transformer blocks. For each $i \in \{1, 2, \ldots, N\}$, the $i$-th transformer block is represented by $\mathcal{T}_i : \mathbb{R}^{n_{i-1}} \times \mathbb{R}^{p_{i-1}} \to \mathbb{R}^{n_i}$, which maps the hidden state $\mathcal{H}_{i-1} \subset \mathbb{R}^{n_{i-1}}$ and parameters $\theta_{i-1} \in \mathbb{R}^{p_{i-1}}$ to the next hidden state $\mathcal{H}_i \subset \mathbb{R}^{n_i}$. Assume that the hidden state $\mathcal{H}_i$ lies on a manifold $\mathcal{N}_i$ with intrinsic dimension $d_i$ embedded in $\mathbb{R}^{n_i}$, while $\mathcal{H}_{i-1}$ lies on a manifold $\mathcal{N}_{i-1}$ with intrinsic dimension $d_{i-1}$ embedded in $\mathbb{R}^{n_{i-1}}$. The rank of the transformer block $\mathcal{T}_i$ is constrained by the inequality*

$$d_i \leq rank(\mathcal{T}_i),$$

*where the rank of $\mathcal{T}_i$ at $\theta_{i-1}$ is defined as $rank(\mathcal{T}_i) = \max_{x \in \mathcal{H}_{i-1}} rank(J(\mathcal{T}_i, x, \theta_{i-1}))$, with $J(\mathcal{T}_i, x, \theta_{i-1})$ representing the Jacobian matrix of $\mathcal{T}_i$ evaluated at $x \in \mathcal{H}_{i-1}$ and $\theta_{i-1}$.*

**Corollary 3.2.1** (**Bound on Parameters for Transformer Block Optimization**). *Let $N_{i-1}$ represent the number of parameters required to optimize at transformer block $i$. Then, the following inequality holds:*

$$\max(d_i - d_{i-1}, 0) \leq N_{i-1}.$$

Recomputing the optimal number of parameters after each gradient step is computationally expensive. However, as training progresses, the model learns to compress data, resulting in fewer parameters being responsible for the local variance of data points. Therefore, it is reasonable to assume that the intrinsic dimensionality of the data and the rank of the transformer blocks decrease during training.

**Conjecture 3.1** (**Transformer Rank Bound Dynamics**). Let $i \in \{1, 2, \ldots, N\}$, and consider the process of fine-tuning. During this process, both the rank of each transformer block $rank(\mathcal{T}_i)$ and the intrinsic dimension $d_i$ of the manifold $\mathcal{H}_i$ decrease. Let $d_i^0$ denote the initial intrinsic dimension. Then, the following inequality holds:

$$d_i^0 \leq \text{rank}(\mathcal{T}_i^t),$$

where $\mathcal{T}_i^t$ represents the transformer block after the $t$-th gradient step. As fine-tuning progresses, this inequality becomes progressively tighter, implying that the gap between the initial intrinsic dimension and the rank of the transformer block reduces over time.

## 3.3 METHODOLOGY

Figure 2 provides a schematic representation of the GeLoRA methodology, which begins by computing the the intrinsic dimensions of data representations across the model's hidden states, allowing for an understanding of the manifold structure that each layer captures. For each layer $i$, let $d_i$ represent the intrinsic dimension of the data manifold at the input, and $d_{i+1}$ the intrinsic dimension at the output. To ensure efficient low-rank adaptation (LoRA) parameters that align with the model's geometry, the minimal rank $r_i$ is set for each layer according to the condition $r_i \geq \max(d_{i+1} - d_i, 0)$, where the difference $d_{i+1} - d_i$ indicates the required capacity to capture any dimensional expansion of the data manifold between consecutive layers. An adaptive scaling factor $\alpha_i$ is then applied across layers to maintain a consistent ratio $\alpha_i/r_i = \text{const}$, preserving the proportion of adaptation strength relative to rank. This enables an efficient fine-tuning process that balances expressivity with computational efficiency.

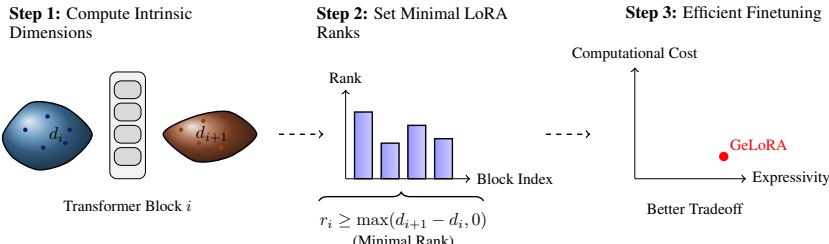

Figure 2: Schematic of the GeLoRA methodology. The process includes intrinsic dimension analysis *(Step 1)*, setting minimal LoRA ranks based on these dimensions *(Step 2)*, and performing efficient fine-tuning to achieve an optimal balance between computational efficiency and model expressivity *(Step 3)*.

To estimate the intrinsic dimension $d_i$ of the hidden state $i$, we employ the two-nearest-neighbors (2-NN) method Facco et al. (2017). Given a dataset in a high-dimensional feature space, we begin by identifying, for each data point $x_j$, its nearest and second-nearest neighbors, computing their respective distances $r_1(j)$ and $r_2(j)$. We then compute the ratio $\mu_j = \frac{r_2(j)}{r_1(j)}$, which encapsulates local geometric information. Under the assumption of locally uniform data density, the cumulative distribution function of the ratio $\mu = \frac{r_2}{r_1}$ is given by

$$F(\mu|d_i) = 1 - \mu^{-d_i}$$

for $\mu \geq 1$ and $d_i > 0$. The intrinsic dimension $d_i$ can be estimated by fitting the empirical distribution of the observed ratios $\{\mu_j\}_{j=1}^N$ to this theoretical distribution, either through maximum likelihood estimation or through linear regression in log-log space of the complementary cumulative distribution.

In high-dimensional settings, the 2-NN method tends to provide a conservative estimate, often serving as a lower bound on the true intrinsic dimension. To illustrate this, we conduct experiments on established benchmark datasets, observing the 2-NN method's behavior relative to the ground truth. To mitigate the risk of underestimating the intrinsic dimension—resulting in an inaccurate value of zero rank in some cases—we add a small offset of 1 to each rank lower bound. Furthermore, rank lower bound is computed for each transformer block as a whole, including the Key, Query, Value, and Output matrices. Since we cannot localize the specific important parameters within each matrix, we set the rank of each matrix in the transformer block equal to the computed intrinsic dimension.

$$r_{K_i} = r_{Q_i} = r_{V_i} = r_{O_i} = \max(d_{i+1} - d_i, 0) + 1,$$

where $r_{K_i}, r_{Q_i}, r_{V_i}$, and $r_{O_i}$ are, respectively, the LoRA ranks of the Key, Query, Value and Output matrices of the transformer block $i$. A pseudocode description of GeLoRA is presented in Appendix B.1.

## 3.4 Fine-Tuning Techniques and Datasets

We evaluate the performance of our GeLoRA technique across several natural language processing tasks. First, we assess its performance on the GLUE benchmark for natural language understanding (Wang et al., 2019), using tasks such as CoLA (Warstadt et al., 2019), SST-2 (Socher et al., 2013), MRPC (Dolan & Brockett, 2005), STS-B (Cer et al., 2017), QNLI (Rajpurkar et al., 2016), and RTE (Dagan et al., 2006; Bar-Haim et al., 2006; Giampiccolo et al., 2007). We then evaluate question answering performance using the SQuAD dataset (Rajpurkar et al., 2016). Finally, we investigate instruction-following tasks by fine-tuning the model on the Airoboros dataset (Durbin, 2024) and evaluating on MT-Bench (Zheng et al., 2023a). For natural language understanding and question answering, we use the DeBERTaV3 model (He et al., 2021), following established practices in the literature. For instruction-following tasks, we fine-tune using Phi-2 (Li et al., 2023). We compare GeLoRA's performance against several fine-tuning techniques, including weight update tuning (Zaken et al., 2022), adapter-based methods (Houlsby et al., 2019; Pfeiffer et al., 2021), and LoRA and its variants (Hu et al., 2021; Ding et al., 2023; Zhang et al., 2023).

## 3.5 Experimental setting

We implemented all algorithms using PyTorch, based on the publicly available HuggingFace Transformers (Wolf et al., 2020) code-base. For optimization, we used the AdamW optimizer (Loshchilov & Hutter, 2019), which features parameters set to $\epsilon = 10^{-6}$, $\beta_1 = 0.9$, and $\beta_2 = 0.999$, and we fixed the batch size to 32. To facilitate fair comparisons across different fine-tuning methods, we employed Optuna (Akiba et al., 2019) for hyperparameter tuning, optimizing parameters such as learning rate, weight decay, warm-up ratio, learning scheduler type, and LoRA dropout over 50 trials for each method. The numerical results were averaged over five runs with random seeds, and we report standard deviations to ensure statistical robustness. The alpha rank ratio for low-rank adaptation techniques was fixed at 32, consistent with prior work (Hu et al., 2021; Zhang et al., 2023), and was not fine-tuned further. For estimating intrinsic dimension, we used the Scikit-Dimension package (Bac et al., 2021). All experiments were conducted on NVIDIA A100-SXM4 GPUs. Additional details regarding the training process can be found in the Appendix E.

## 3.6 Numerical Results

### 3.6.1 Natural Language Understanding: GLUE Benchmark

Table 1: Results with DeBERTaV3-base on GLUE test set. The best results for each dataset are highlighted in **bold**, while the second-best results are underlined. We report the average correlation for STS-B. *Full FT* represent full fine-tuning, *HA Adapter* represents Houlsby Adapters, and *PF Adapter* represents Pfeiffer Adapters.

| Method | # Params | CoLA | STS-B | MRPC | QNLI | SST-2 | RTE | QQP | MNLI | Average |
|---|---|---|---|---|---|---|---|---|---|---|
| Full FT | 184.42M | 68.28 ± 1.39 | 91.32 ± 0.45 | 73.53 ± 3.25 | 93.81 ± 0.21 | 94.68 ± 0.30 | 73.67 ± 1.33 | 88.54 ± 0.23 | 89.65 ± 0.19 | 84.19 |
| BitFit | 0.11M | 68.66 ± 1.87 | 89.40 ± 0.57 | 85.2 ± 1.56 | 92.10 ± 0.13 | 94.54 ± 0.30 | 75.11 ± 2.52 | 86.25 ± 0.27 | 86.04 ± 0.58 | 84.66 |
| HA Adapter | 0.65M | 68.46 ± 1.08 | 91.26 ± 0.13 | 86.76 ± 0.44 | 93.52 ± 0.40 | 95.32 ± 0.35 | 80.43 ± 2.78 | 89.08 ± 0.06 | 89.07 ± 0.19 | 86.74 |
| PF Adapter | 0.62M | 68.59 ± 1.43 | 89.85 ± 0.13 | 88.24 ± 1.07 | 93.33 ± 0.30 | 95.55 ± 0.41 | 79.14 ± 2.95 | 88.60 ± 0.14 | 88.82 ± 0.07 | 86.52 |
| LoRA$_{r=1}$ | 0.08M | 69.68 ± 0.92 | 88.29 ± 3.28 | 88.43 ± 1.37 | 93.83 ± 0.13 | 95.04 ± 0.43 | 80.29 ± 1.33 | 90.41 ± 0.05 | 89.64 | 86.95 |
| LoRA$_{r=2}$ | 0.15M | 69.04 ± 1.51 | 88.60 ± 3.09 | 87.75 ± 0.69 | 93.79 ± 0.17 | 95.04 ± 0.22 | 80.43 ± 1.60 | 90.78 ± 0.11 | 89.77 | 86.90 |
| SoRA$_{r=1}$ | 0.08M | 61.78 ± 2.37 | 78.88 ± 6.55 | 87.45 ± 3.06 | 88.66 ± 0.68 | 91.94 ± 0.52 | 82.32 ± 2.49 | | | 81.17 |
| SoRA$_{r=2}$ | 0.15M | 67.85 ± 1.33 | 84.33 ± 3.90 | 88.04 ± 2.00 | 89.76 ± 0.41 | 91.40 ± 0.32 | 78.84 ± 3.74 | | | 84.04 |
| AdaLoRA$_{r=1}$ | 0.15M | 69.28 ± 0.33 | 92.08 ± 0.15 | 84.61 ± 0.91 | 93.84 ± 0.15 | 95.07 ± 0.42 | 74.96 ± 3.82 | 89.92 ± 0.10 | 90.12 ± 0.20 | 86.23 |
| AdaLoRA$_{r=2}$ | 0.22M | 64.76 ± 1.49 | 91.56 ± 0.12 | 87.25 ± 0.93 | 94.07 ± 0.12 | 95.44 ± 0.34 | 81.87 ± 0.95 | 90.12 ± 0.08 | 90.13 ± 0.26 | 86.90 |
| GeLoRA | – | 70.96 ± 0.96 | 91.66 ± 0.48 | 89.9 ± 0.79 | 93.87 ± 0.23 | 95.05 ± 0.24 | 81.29 ± 1.64 | 90.81 ± 0.12 | 89.84 ± 0.22 | 87.92 |

Our experimental results demonstrate the effectiveness of GeLoRA across multiple tasks in the GLUE benchmark. As shown in Table 1, GeLoRA achieves competitive or superior performance compared to existing parameter-efficient fine-tuning methods while maintaining a minimal parameter footprint. Specifically, GeLoRA obtains an average score of 87.92 across all evaluated tasks, outperforming strong baselines like HA Adapter (86.74), LoRA (86.95) and its variants (86.90). On individual tasks, GeLoRA shows particularly strong performance on CoLA (70.96) and MRPC (89.90), achieving the best results among all parameter-efficient methods, while maintaining competitive performance on other tasks. The results are particularly impressive when considering the performance-to-parameter ratio. While other techniques achieves better results on some tasks (e.g.,

95.55 on SST-2), it requires six orders of magnitude more parameters. Our method maintains comparable performance while being substantially more parameter-efficient, making it particularly suitable for resource-constrained scenarios. What's particularly noteworthy is GeLoRA's parameter efficiency, as detailed in Table 2. The method adaptively allocates parameters based on task complexity, ranging from 0.09M parameters for simpler tasks like QNLI and SST-2, to 0.13M parameters for more complex tasks such as MRPC and RTE. This adaptive parameter allocation results in optimal mean ranks while using significantly fewer parameters compared to full fine-tuning (184.42M parameters) and competitive with other efficient methods like LoRA and its variants (0.08M-0.22M parameters). Furthermore, our approach is more intuitive because models do not need to treat all datasets or tasks equally. During the pretraining phase, they may have already gained prior knowledge relevant to certain tasks or datasets, which reduces the need for extensive fine-tuning to achieve strong performance.

Table 2: Number of parameters in GeLoRA for each task.

| Task | CoLA | STS-B | MRPC | QNLI | SST-2 | RTE | MNLI | QQP |
|------|------|-------|------|------|-------|-----|------|-----|
| # Params | 0.10M | 0.11M | 0.13M | 0.09M | 0.09M | 0.13M | 0.10M | 0.12M |
| Mean Rank | 1.33 | 1.50 | 1.75 | 1.25 | 1.17 | 1.75 | 1.33 | 1.58 |
| Rounded Mean Rank | 1 | 2 | 2 | 1 | 1 | 2 | 1 | 2 |

A potential question that arises is how increasing the LoRA ranks uniformly, or introducing greater complexity into the adaptive variants AdaLoRA and SoRA, might impact their performance, and how they would compare to GeLoRA. To address this, we conduct a comparison in a high-rank setting, where we adjust the lower rank bounds of GeLoRA by applying an offset to align with the higher ranks selected for the other fine-tuning techniques. Specifically, we set the ranks as follows:

$$r_{K_i} = r_{Q_i} = r_{V_i} = r_{O_i} = \max(d_{i+1} - d_i, 0) + o,$$

where $o$ is the applied offset to GeLoRA ranks.

Table 3: Results with DeBERTaV3-base on GLUE test set using higher ranks. The best results for each dataset are highlighted in **bold**, while the second-best results are underlined. We report the average correlation for STS-B.

| Method | # Params | CoLA | STS-B | MRPC | QNLI | SST-2 | RTE | Average |
|--------|----------|------|-------|------|------|-------|-----|---------|
| LoRA$_{r=4}$ | 0.3M | $67.52 \pm 0.38$ | $89.84 \pm 1.36$ | $\mathbf{89.12 \pm 2.09}$ | $\underline{93.77 \pm 0.10}$ | $95.39 \pm 0.40$ | $81.73 \pm 1.55$ | $\underline{86.23}$ |
| SoRA$_{r=4}$ | 0.3M | $63.47 \pm 1.99$ | $81.68 \pm 7.93$ | $87.06 \pm 1.15$ | $90.04 \pm 0.67$ | $92.46 \pm 0.59$ | $\mathbf{86.09 \pm 2.69}$ | $83.47$ |
| AdaLoRA$_{r=4}$ | 0.44M | $\mathbf{68.62 \pm 1.22}$ | $90.54 \pm 0.23$ | $84.31 \pm 1.45$ | $\mathbf{94.11 \pm 0.12}$ | $95.39 \pm 0.44$ | $79.71 \pm 1.24$ | $85.45$ |
| GeLoRA | $--$ | $\underline{68.53 \pm 0.71}$ | $\mathbf{91.38 \pm 0.43}$ | $\underline{88.12 \pm 0.73}$ | $93.15 \pm 0.17$ | $\mathbf{95.44 \pm 0.35}$ | $80.92 \pm 1.47$ | $\mathbf{86.26}$ |

Moreover, using dataset-specific ranks aligns with a common practice used to enhance performance during fine-tuning across various benchmarks, which is intermediate task tuning. This approach involves fine-tuning a model on a different task from the target task as a preliminary warm-up step. While this methodology is primarily intuitively motivated—rooted in the idea of learning common features and fostering common-sense reasoning—its theoretical justification remains less clear. In this regard, we aim to provide a plausible explanation for the effectiveness of this approach. We focus on three tasks: MRPC (Dolan & Brockett, 2005), STS-B (Cer et al., 2017), and RTE (Dagan et al., 2006; Bar-Haim et al., 2006; Giampiccolo et al., 2007). Although each dataset has a specific focus, they all assess semantic relationships between pairs of texts, presenting a strong case for a sequential fine-tuning strategy. MRPC targets the identification of paraphrases, where two sentences convey the same idea using different wording. STSB evaluates the degree of semantic similarity between sentences on a continuous scale ranging from 0 to 5. RTE determines whether one sentence entails another, reflecting a distinct aspect of semantic relationships. These tasks require the model to comprehend nuanced semantic properties, including synonyms, paraphrases, and entailment. As a result, the underlying language representations across these datasets exhibit significant similarities. Consequently, we hypothesize that fine-tuning on MRPC can facilitate the subsequent fine-tuning processes for STSB and RTE.

We posit that the main reason for this enhanced performance stems from data compression, as the model learns features relevant to the target tasks during intermediate training. To evaluate this hypothesis, we theorize that the lower bound of intrinsic dimensions will become looser after compression. Our experimental results support this hypothesis. For instance, we observe a decrease in

the mean intrinsic dimension for RTE (from 13.47 to 12.97) as shown in Figure 3a, while Figure 3b shows that the mean intrinsic dimension for STS-B remains consistent (from 13.19 to 13.01), albeit with a change in their profiles as shown in Figure 3a and Figure 3b. Additionally, we note similarities in the behavior of different layers: the lower layers, responsible for basic features (such as syntax and grammar), remain largely unchanged; however, the higher layers, which capture more complex features, exhibit significant compression. The intermediate layers, as indicated by recent studies on the geometry of hidden representations, show a slight increase in their capacity due to the model's specialization in the semantics of the intermediate task. Thus, the decrease in the mean intrinsic dimensions corresponds to a reduction in the lower bounds presented in Corollary 3.2.1. This loosening of the bounds indicates that the number of parameters required for optimal performance has decreased, leading to more efficient training.

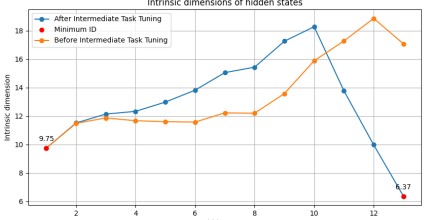
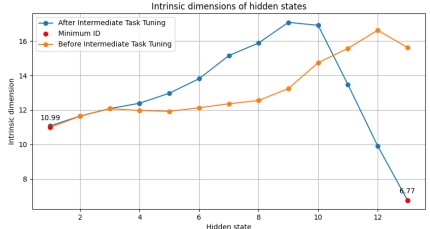

(a) Intrinsic dimension profile of the RTE dataset using DebertaV3 before and after intermediate task tuning using MRPC.

(b) Intrinsic dimension profile of the STS-B dataset using DebertaV3 before and after intermediate task tuning using MRPC.

Figure 3: Intrinsic dimension profiles of RTE and STS-B datasets using DebertaV3 before and after intermediate task tuning using MRPC.

Finally, we evaluate the efficiency of different techniques pertaining to the same budget constraint. We measured the clock time for training across six datasets, conducting experiments for 20 epochs on all datasets except for RTE, which was run for 50 epochs. All experiments were executed on identical computing infrastructure, using eight NVIDIA A100-SXM4 GPUs with a consistent batch size of 32. To ensure a fair comparison between different techniques, we adjusted the ranks of LoRA and its variants to match the rounded mean rank of GeLoRA.

Table 4 reveals that GeLoRA demonstrates superior performance while incurring less computational overhead compared to the other techniques. In contrast, the SoRA method experiences additional computational overhead during training due to the gradient calculations required for the proximal gradient approach used to enforce sparsity via the $l_0$ norm. On the other hand, BitFit requires training the task-specific head for better performance which adds complexity to the method.

Table 4: Training computational cost (runtime) in seconds for DeBERTaV3-base fine-tuning on GLUE tasks. The runtime for each fine-tuning is indicated in seconds. The best results for each dataset are highlighted in **bold**.

| Dataset | GeLoRA | SoRA | LoRA | AdaLoRA | BitFit | HAdapter | PAdaper |
|---------|--------|------|------|---------|--------|----------|---------|
| CoLA | $\mathbf{85.68 \pm 2.27}$ | $159.42 \pm 0.80$ | $100.95 \pm 10.53$ | $165.43 \pm 0.28$ | $157.27 \pm 1.07$ | $117.98 \pm 0.07$ | $113.52 \pm 0.11$ |
| STS-B | $\mathbf{59.13 \pm 3.26}$ | $116.19 \pm 0.50$ | $78.26 \pm 6.92$ | $157.50 \pm 8.36$ | $122.68 \pm 0.40$ | $84.51 \pm 0.06$ | $81.27 \pm 0.04$ |
| MRPC | $\mathbf{40.42 \pm 0.30}$ | $86.03 \pm 1.13$ | $58.75 \pm 1.73$ | $112.61 \pm 1.36$ | $94.93 \pm 0.34$ | $57.41 \pm 0.10$ | $55.09 \pm 0.03$ |
| QNLI | $\mathbf{736.57 \pm 3.34}$ | $1617.92 \pm 1.94$ | $865.76 \pm 4.11$ | $2328.60 \pm 24.81$ | $1341.47 \pm 21.03$ | $1254.14 \pm 1.21$ | $1205.86 \pm 1.83$ |
| SST-2 | $\mathbf{475.58 \pm 5.10}$ | $1041.62 \pm 1.82$ | $482.38 \pm 5.11$ | $1140.65 \pm 2.25$ | $871.10 \pm 5.05$ | $807.91 \pm 0.57$ | $775.33 \pm 0.56$ |
| RTE | $\mathbf{75.62 \pm 0.29}$ | $158.25 \pm 1.43$ | $116.28 \pm 7.30$ | $207.89 \pm 4.42$ | $80.5 \pm 0.24$ | $104.38 \pm 0.06$ | $100.40 \pm 0.11$ |
| Average | $\mathbf{245.5}$ | $529.91$ | $283.73$ | $685.45$ | $444.66$ | $404.39$ | $388.58$ |

### 3.6.2 QUESTION ANSWERING: SQUAD

Our experimental results demonstrate the efficiency of GeLoRA against baseline approaches on SQuADv1.1 and SQuADv2.0 benchmarks. GeLoRA achieves state-of-the-art performance with EM/F1 scores of 86.72/92.84 and 83.15/86.25 respectively, surpassing other fine-tuning techniques

while using only a fraction of trainable parameters. Table 5 reveals consistent performance improvements over existing parameter-efficient methods. GeLoRA outperforms LoRA variants by margins of 0.45-2.27 points in EM score on SQuADv1.1, with similar gains observed on SQuADv2.0. The performance delta is more pronounced when compared to adapter-based methods, showing improvements of 2.14 and 3.72 points over HAdapter and PAdapter respectively on the SQuAD v1.1 dataset.

Table 5: Results with DeBERTaV3-base on SQuADv1.1 and SQuADv2.0. Here # Params is the number of trainable parameters. We report both the exact match and F1-score. The best results in each setting are shown in bold.

| | # Params | SQuADv1.1 | | SQuADv2.0 | |
|---|---|---|---|---|---|
| | | EM | F1 | EM | F1 |
| Full FT | 183.83M | $86.12 \pm 0.28$ | $92.68 \pm 0.13$ | $83.03 \pm 0.49$ | $86.21 \pm 0.51$ |
| HAdapter | 0.06M | $84.58 \pm 0.20$ | $91.57 \pm 0.13$ | $80.79 \pm 1.10$ | $84.28 \pm 1.13$ |
| PAdapter | 0.03M | $83.00 \pm 0.06$ | $90.57 \pm 0.10$ | $78.17 \pm 0.95$ | $81.94 \pm 0.94$ |
| $\text{LoRA}_{r=2}$ | 0.01M | $84.45 \pm 0.35$ | $91.35 \pm 0.25$ | $\mathbf{83.15 \pm 0.77}$ | $86.16 \pm 0.74$ |
| $\text{LoRA}_{r=1}$ | $7e^{-3}$M | $86.23 \pm 0.16$ | $92.51 \pm 0.16$ | $81.09 \pm 0.66$ | $84.22 \pm 0.63$ |
| $\text{AdaLoRA}_{r=1}$ | 0.15M | | | $81.12 \pm 0.35$ | $84.23 \pm 0.31$ |
| $\text{AdaLoRA}_{r=2}$ | 0.22M | $86.27 \pm 0.31$ | $92.61 \pm 0.27$ | $81.68 \pm 0.51$ | $84.80 \pm 0.50$ |
| GeLoRA | $8e^{-3}$M | $\mathbf{86.72 \pm 0.27}$ | $\mathbf{92.84 \pm 0.20}$ | $\mathbf{83.15 \pm 0.22}$ | $\mathbf{86.25 \pm 0.24}$ |

## 4  CONCLUSION AND FUTURE WORK

In this work, we introduced GeLoRA, a theoretically grounded technique designed for the efficient fine-tuning of large language models. GeLoRA effectively addresses the expressivity-efficiency trade-off inherent in low-rank adaptation techniques. Our approach is straightforward yet powerful, supported by theoretical analyses that ensure an optimal balance between expressivity and computational efficiency. We theoretically demonstrated that the number of parameters requiring optimization per transformer block is lower bounded by the difference in the intrinsic dimensions of the corresponding input and output hidden representations. This finding provides a method for estimating the optimal ranks for low-rank adaptation techniques, and connecting the manifold of data representations to the manifold of model parameters. Empirically, our methodology surpasses current state-of-the-art approaches on the GLUE benchmarks while maintaining computational efficiency. Additionally, GeLoRA offers a potential theoretical justification for the effectiveness of intermediate task tuning in certain scenarios. However, we acknowledge that our technique shifts some computational overhead to the preprocessing step and relies on a local estimator of intrinsic dimensions, specifically the Two Nearest Neighbors (TwoNN) method. We believe this aspect can be further improved through the application of persistent homology dimensions to estimate intrinsic dimensions, as this approach considers both local and global topological features of the manifold. Moreover, it can be computed efficiently on GPUs by leveraging parallelization.

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

# A    MATHEMATICAL FORMALISM

In this section, we provide the mathematical definitions, theorems, and algorithms that serve as the foundation for the methods and theorems presented in this paper.

## A.1    INTRINSIC DIMENSIONALITY

### A.1.1    DEFINITION AND EXAMPLE

**Definition A.1** (**Intrinsic Dimensionality**). Let $\mathcal{M} \subseteq \mathbb{R}^D$ be a manifold embedded in a $D$-dimensional ambient space. The *intrinsic dimensionality (ID)* of $\mathcal{M}$ is defined as the smallest number of coordinates $d$ such that all data points on $\mathcal{M}$ can be locally approximated by a $d$-dimensional Euclidean space. Formally, for every point $\mathbf{x} \in \mathcal{M}$, there exists a neighborhood $\mathcal{N}(\mathbf{x})$ and a smooth map $\phi : \mathbb{R}^d \to \mathbb{R}^D$ such that $\phi(\mathbb{R}^d) \cap \mathcal{N}(\mathbf{x}) = \mathcal{M} \cap \mathcal{N}(\mathbf{x})$.

In practical terms, the intrinsic dimensionality $d$ represents the number of degrees of freedom required to describe the structure of $\mathcal{M}$, regardless of the ambient space's dimensionality $D$.

**Example.**    Consider a helical curve $\mathcal{H}$ embedded in three-dimensional space ($\mathbb{R}^3$) (Figure 4) with the parametric representation:

$$\mathbf{x}(t) = \begin{bmatrix} r\cos(t) \\ r\sin(t) \\ ct \end{bmatrix}, \quad t \in \mathbb{R},$$

where $r > 0$ is the radius and $c > 0$ is the vertical scaling factor.

- Although the helix is embedded in $\mathbb{R}^3$, the parameter $t$ uniquely determines any point on the curve.

- Hence, the helix is a one-dimensional ($d = 1$) manifold, since it can be locally approximated by a one-dimensional Euclidean space ($\mathbb{R}^1$).

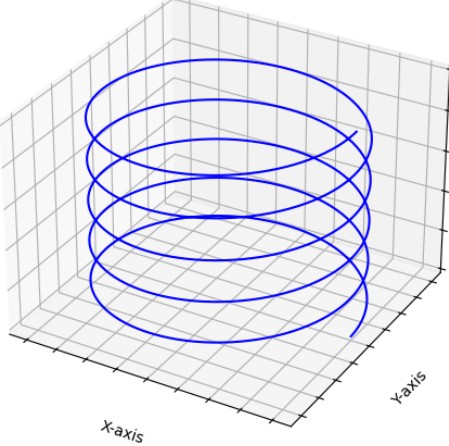

Figure 4: A helical curve in 3D space with an intrinsic dimension of 1, fully described by a single parameter despite its 3D embedding.

### A.1.2    METHODOLOGY: TWO NEAREST NEIGHBORS ESTIMATOR

**Choice Justification.**    The *Two Nearest Neighbor* (TwoNN) intrinsic dimension (ID) estimator, proposed by Facco et al. (2017), uses local geometric properties to estimate the intrinsic dimension of datasets. It is particularly well-suited for NLP datasets because:

- **Scalability:** The estimator efficiently computes the ID using only distances to the first two nearest neighbors of each point, even for large datasets.

- **Robustness:** It produces consistent results across dataset scales Denti et al. (2022).

- **Validity:** The assumption of local constant density, crucial for the TwoNN method, is satisfied in NLP datasets, as validated using the Point Adaptive kNN (PAk) method (Rodriguez et al., 2018; Valeriani et al., 2023).

**Methodology.** The TwoNN estimator follows these steps:

1. **Nearest Neighbor Distances:** For each data point $x_i$, compute:
   - $r_{i1}$: Distance to its first nearest neighbor.
   - $r_{i2}$: Distance to its second nearest neighbor.

2. **Compute Ratios:** Calculate the ratio $\mu_i = \frac{r_{i2}}{r_{i1}}$ for each point.

3. **Pareto Distribution Assumption:** The ratios $\mu_i$ follow a Pareto distribution $p(\mu_i | d) = d\mu_i^{-d-1}$, where $d$ is the intrinsic dimension.

4. **Cumulative Distribution Function (CDF):** The Pareto CDF is given by $F(\mu) = 1 - \mu^{-d}$. The empirical CDF is approximated as:

$$F_{\text{emp}}(\mu_{\sigma(i)}) = \frac{i}{N},$$

   where $\mu_{\sigma(i)}$ are the sorted values of $\mu_i$ in ascending order and $N$ is the total number of points.

5. **Linear Regression:** By plotting $\log(\mu_{\sigma(i)})$ against $-\log(1 - F_{\text{emp}}(\mu_{\sigma(i)}))$, the slope of the line gives the intrinsic dimension $d$.

To ensure robustness, the TwoNN estimator is applied to random subsets of the dataset of decreasing sizes (e.g., $N, N/2, N/4, \dots$), and the ID is chosen where the estimates stabilize.

**Hands-on Example: Manual TwoNN Calculation.** Consider a toy dataset of five points in two dimensions, with coordinates:

$$x_1 = (0,0),\ x_2 = (1,0),\ x_3 = (2,0),\ x_4 = (0,1),\ x_5 = (2,2).$$

- **Step 1: Compute Nearest Neighbor Distances** For each point $x_i$, compute distances to all other points. For example:

$$\text{For } x_1 : r_{12} = 1,\ r_{13} = 2,\ r_{14} = 1,\ r_{15} \approx 2.83.$$

  The nearest neighbors of $x_1$ are $r_{i1} = 1$ and $r_{i2} = 1$.

- **Step 2: Compute Ratios** For each point $x_i$, calculate $\mu_i = \frac{r_{i2}}{r_{i1}}$. For example:

$$\mu_1 = \frac{r_{i2}}{r_{i1}} = \frac{2}{1} = 2, \quad \mu_2 \approx 1.41.$$

- **Step 3: Sort Ratios and Compute Empirical CDF** Sort the ratios $\mu_i$ in ascending order:

$$\mu_{\sigma(1)} \leq \mu_{\sigma(2)} \leq \cdots \leq \mu_{\sigma(N)}.$$

  The empirical CDF is:

$$F_{\text{emp}}(\mu_{\sigma(i)}) = \frac{i}{N}.$$

- **Step 4: Linear Regression to Estimate $d$** Take the logarithm of the sorted ratios:

$$\log(\mu_{\sigma(i)}), \quad \text{and } -\log(1 - F_{\text{emp}}(\mu_{\sigma(i)})).$$

  Plot $\log(\mu_{\sigma(i)})$ vs. $-\log(1 - F_{\text{emp}}(\mu_{\sigma(i)}))$ and fit a straight line through the origin. The slope of the line is the intrinsic dimension $d$.

**Computational Complexity.** The TwoNN estimator requires finding the two nearest neighbors for each data point in the dataset. This operation has a computational complexity of $O(n^2)$ for a naive approach or $O(n \log(n))$ when using optimized nearest neighbor search methods (e.g., KD-trees or ball trees). For a dataset with $n$ points and a model with $L$ transformer blocks (e.g., where distances need to be computed across $L$ hidden representations), the overall complexity becomes:

$$O(L \cdot n \log(n)) \quad \text{or} \quad O(L \cdot n^2),$$

depending on the algorithm used for nearest neighbor computation.

### A.2 TRANSFORMER ARCHITECTURE

**Definition A.2** (**Single-head Self-attention Layer**). Let $k, d \in \mathbb{N}$. Consider matrices $Q, K, V \in \mathbb{R}^{k \times d}$. For any integer $n \in \mathbb{N}$ and vectors $x_1, \ldots, x_n \in \mathbb{R}^d$, self-attention with parameters $(Q, K, V)$ maps the sequence $(x_1, \ldots, x_n) \in \mathbb{R}^{d \times n}$ to

$$f(x_1, \ldots, x_n) = \left( V \sum_{j=1}^{n} \text{softmax} \left( \frac{x_i^\top Q^\top K x_j}{\sqrt{k}} \right) x_j \right)_{1 \leq i \leq n} \in (\mathbb{R}^k)^n, \tag{1}$$

**Definition A.3** (**Multi-head Self Attention Layer**). Let $d \in \mathbb{N}$ and $H$ be a divisor of $d$. For $1 \leq h \leq H$, let $Q(h), K(h), V(h) \in \mathbb{R}^{k \times d}$ with $k := d/H$, and $W(h) \in \mathbb{R}^{d \times k}$. Multi-head self-attention with parameters $(Q(h), K(h), V(h), W(h))_{1 \leq h \leq H}$ maps any sequence $(x_1, \ldots, x_n) \in (\mathbb{R}^d)^n$ to

$$f_{\text{MH}}(x_1, \ldots, x_n) = \sum_{h=1}^{H} W(h) f^{(h)}(x_1, \ldots, x_n) \in (\mathbb{R}^d)^n, \tag{2}$$

where $f^{(h)}$ denotes single-head self-attention with parameters $(Q(h), K(h), V(h))$.

## B GELORA: FRAMEWORK AND THEORETICAL PROOFS

In this section, we provide the pseudocode for the GeLoRA framework along with detailed proofs of the theorems presented in this paper.

## B.1 GeLoRA: Pseudocode

---

**Algorithm 1** GeLoRA (Geometry-aware Low-Rank Adaptation)

---

**Require:**
1: Model $\mathcal{M}$ with $L$ transformer layers
2: Dataset $\mathcal{D}$
3: Desired constant ratio $c$ for $\alpha_i/r_i$

**Ensure:**
4: Optimal LoRA ranks $r_i$ and scaling factors $\alpha_i$ for each layer
5: **function** EstimateIntrinsicDimension($\mathbf{X}$)
6:     **for** each point $x_j$ in $\mathbf{X}$ **do**
7:         $r_1(j) \leftarrow$ distance to nearest neighbor of $x_j$
8:         $r_2(j) \leftarrow$ distance to second nearest neighbor of $x_j$
9:         $\mu_j \leftarrow r_2(j)/r_1(j)$
10:     **end for**
11:     // Fit empirical distribution to theoretical CDF: $F(\mu|d) = 1 - \mu^{-d}$
12:     $d \leftarrow$ FitDistribution($\{\mu_j\}$)         ▷ Using log-log regression or MLE
13:     **return** $d$
14: **end function**
15: **function** ComputeGeLoRAParameters($\mathcal{M}, \mathcal{D}$)
16:     // Initialize arrays for dimensions and parameters
17:     $\mathbf{d} \leftarrow$ array of size $L+1$         ▷ Intrinsic dimensions
18:     $\mathbf{r} \leftarrow$ array of size $L$         ▷ LoRA ranks
19:     $\boldsymbol{\alpha} \leftarrow$ array of size $L$         ▷ Scaling factors
20:     // Step 1: Compute intrinsic dimensions for each layer
21:     **for** $i \leftarrow 0$ to $L$ **do**
22:         $\mathbf{X}_i \leftarrow$ GetHiddenStates($\mathcal{M}, \mathcal{D}, \text{layer} = i$)
23:         $d_i \leftarrow$ EstimateIntrinsicDimension($\mathbf{X}_i$)
24:     **end for**
25:     // Step 2: Compute ranks for each layer
26:     **for** $i \leftarrow 0$ to $L-1$ **do**
27:         dim_difference $\leftarrow \max(d_{i+1} - d_i, 0)$
28:         base_rank $\leftarrow$ dim_difference $+ 1$         ▷ Add offset of 1
29:         // Set equal ranks for all matrices in transformer block
30:         $r_{K_i}, r_{Q_i}, r_{V_i}, r_{O_i} \leftarrow$ base_rank
31:     **end for**
32:     // Step 3: Compute scaling factors maintaining $\alpha_i/r_i = $ const
33:     total_rank $\leftarrow \sum_{i=0}^{L-1} r_i$
34:     **for** $i \leftarrow 0$ to $L-1$ **do**
35:         $\alpha_i \leftarrow c \cdot r_i$         ▷ Ensures $\alpha_i/r_i = c$
36:     **end for**
37:     **return** $\mathbf{r}, \boldsymbol{\alpha}$
38: **end function**
39: // Main execution
40: **function** Main
41:     model $\leftarrow$ LoadModel()
42:     dataset $\leftarrow$ LoadDataset()
43:     ranks, scaling_factors $\leftarrow$ ComputeGeLoRAParameters(model, dataset)
44:     ApplyLoRAParameters(model, ranks, scaling_factors)
45: **end function**

---

## B.2 MATHEMATICAL PROOFS

### B.2.1 PROOF OF THEOREM 3.1 – INTRINSIC DIMENSION AS A LOWER BOUND

**Theorem B.1 (Intrinsic Dimension as a Lower Bound).** *The intrinsic dimension $\hat{idim}(\phi)$ is a lower bound to the local dimensionality $d(\phi)$.*

$$d(\phi) \geq \hat{idim}(\phi).$$

*Proof.* The local dimensionality $d(\phi)$ of a neuromanifold is defined as the rank of the Fisher Information Matrix (FIM), which corresponds to the number of non-zero eigenvalues of the FIM. However, in practice, while the FIM is almost surely of full rank, many of its eigenvalues can be exceedingly small, on the order of $\epsilon \in \mathbb{R}^+$, where $\epsilon$ is a small positive threshold.

According to the Cramér-Rao bound, the variance of parameter estimates is inversely proportional to the eigenvalues of the FIM. Specifically, for an eigenvalue on the order of $\epsilon$, the variance of the corresponding parameter is at least $1/\epsilon$. Parameters associated with such small eigenvalues contribute negligible information about the model and can therefore be considered effectively uninformative.

By disregarding parameters associated with small eigenvalues, we obtain the definition of the intrinsic dimension $\text{idim}(\phi)$, which represents the minimal number of parameters necessary to describe the structure of the manifold. The specific value of the intrinsic dimension depends on the threshold $\epsilon$ used to exclude eigenvalues below a certain magnitude. This threshold determines the uninformative directions that are discarded, yielding an estimate of the ground truth intrinsic dimension. Consequently, the estimated intrinsic dimension $\hat{idim}(\phi)$ is always less than or equal to the local dimensionality $d(\phi)$:

$$\hat{idim}(\phi) \leq d(\phi).$$

Thus, the estimated intrinsic dimension $\hat{idim}(\phi)$ provides a lower bound for the local dimensionality $d(\phi)$, completing the proof. $\square$

### B.2.2 PROOF OF THEOREM 3.2 – RANK BOUND OF TRANSFORMER BLOCKS

**Theorem B.2 (Rank Bound of Transformer Blocks).** *Let $\mathcal{M}$ denote a language model consisting of $N$ transformer blocks. For each $i \in \{1, 2, \ldots, N\}$, the $i$-th transformer block is represented by $\mathcal{T}_i : \mathbb{R}^{n_{i-1}} \times \mathbb{R}^{p_{i-1}} \to \mathbb{R}^{n_i}$, which maps the hidden state $\mathcal{H}_{i-1} \subset \mathbb{R}^{n_{i-1}}$ and parameters $\theta_{i-1} \in \mathbb{R}^{p_{i-1}}$ to the next hidden state $\mathcal{H}_i \subset \mathbb{R}^{n_i}$. Assume that the hidden state $\mathcal{H}_i$ lies on a manifold $\mathcal{N}_i$ with intrinsic dimension $d_i$ embedded in $\mathbb{R}^{n_i}$, while $\mathcal{H}_{i-1}$ lies on a manifold $\mathcal{N}_{i-1}$ with intrinsic dimension $d_{i-1}$ embedded in $\mathbb{R}^{n_{i-1}}$. The rank of the transformer block $\mathcal{T}_i$ is constrained by the inequality*

$$d_i \leq rank(\mathcal{T}_i),$$

*where the rank of $\mathcal{T}_i$ at $\theta_{i-1}$ is defined as $rank(\mathcal{T}_i) = \max_{x \in \mathcal{H}_{i-1}} rank(J(\mathcal{T}_i, x, \theta_{i-1}))$, with $J(\mathcal{T}_i, x, \theta_{i-1})$ representing the Jacobian matrix of $\mathcal{T}_i$ evaluated at $x \in \mathcal{H}_{i-1}$ and $\theta_{i-1}$.*

*Proof.* Let $i \in \{1, 2, \ldots, N\}$, and consider the map $\mathcal{T}_i : \mathbb{R}^{n_{i-1}} \times \mathbb{R}^{p_{i-1}} \to \mathbb{R}^{n_i}$ to be the $i$-th transformer block, which maps the hidden state $\mathcal{H}_{i-1} \subset \mathbb{R}^{n_{i-1}}$ and parameters $\theta_{i-1} \in \mathbb{R}^{p_{i-1}}$ to the next hidden state $\mathcal{H}_i \subset \mathbb{R}^{n_i}$. Assume that $\text{idim}(\mathcal{H}_{i-1}) = d_{i-1}$ and $\text{idim}(\mathcal{H}_i) = d_i$.

Given that $\text{idim}(\mathcal{H}_{i-1}) = d_{i-1} \leq n_{i-1}$, we can define a smooth bijective parameterization $\phi : \mathcal{U} \to \mathbb{R}^{n_{i-1}}$ from an open set $\mathcal{U} \subset \mathbb{R}^{d_{i-1}}$ to an open subset $\mathcal{O} \subset \mathcal{H}_{i-1}$. We now extend this parameterization to include the parameters $\theta_{i-1} \in \mathbb{R}^{p_{i-1}}$ by considering the map $\psi : \mathcal{U} \to \mathbb{R}^{n_{i-1}} \times \mathbb{R}^{p_{i-1}}$ that maps each point $x \in \mathcal{U}$ to $(x, \theta_{i-1})$.

Since $\mathcal{T}_i$ is smooth almost everywhere, we can apply the constant rank theorem [1] for manifolds to the composed map $\mathcal{T}_i \circ \psi$, obtaining:

$$\text{idim}(\mathcal{T}_i(\mathcal{H}_{i-1})) = \text{rank}(\mathcal{T}_i \circ \psi) = \text{rank}(J_{\mathcal{T}_i \circ \psi}),$$

---

[1] By Sard's Theorem (Guillemin & Pollack, 2010), critical points—where the Jacobian rank is lower—map to a set of measure zero. These regions of lower ranks contribute negligibly to the representation manifolds. Therefore, we can disregard them and focus only on regions where the rank is constant and maximal.

where $J_{\mathcal{T}_i \circ \psi}$ is the Jacobian matrix of the composition $\mathcal{T}_i \circ \psi$.

Using the chain rule, the rank of the composition is bounded by the minimum rank of the individual Jacobians:

$$\text{rank}(J_{\mathcal{T}_i \circ \psi}) = \text{rank}(J_{\mathcal{T}_i} \cdot J_\psi) \leq \text{rank}(J_{\mathcal{T}_i})$$

Thus, the dimension of $\mathcal{T}_i(\mathcal{H}_{i-1})$, which corresponds to the intrinsic dimension $d_i$ of the hidden state $\mathcal{H}_i$, satisfies:

$$d_i = \text{idim}(\mathcal{H}_i) \leq \text{rank}(\mathcal{T}_i).$$

This completes the proof. $\qquad\square$

### B.2.3 Proof of Corollary 3.2.1 – Bound on Parameters for Transformer Block Optimization

**Corollary B.2.1** (**Bound on Parameters for Transformer Block Optimization**). *Let $N_{i-1}$ represent the number of parameters required to optimize at transformer block $i$. Then, the following inequality holds:*

$$\max(d_i - d_{i-1}, 0) \leq N_{i-1}.$$

*Proof.* We begin by considering the result from Theorem 3.2, which asserts:

$$d_i \leq \text{rank}(\mathcal{T}_i).$$

where $\mathcal{T}_i$ is the transformation applied at block $i$.

The rank of $\mathcal{T}_i$, $\text{rank}(\mathcal{T}_i)$, corresponds to the number of non-noisy directions in its input space, meaning $\theta_i$ and $x \in \mathcal{H}_{i-1}$.

$$\text{rank}(\mathcal{T}_i) = \text{Number of non-noisy directions at } \mathcal{H}_{i-1} + \text{Number of non-noisy directions at } \theta_i$$

By the definition of intrinsic dimensionality, the number of non-noisy directions at $\mathcal{H}_{i-1}$ is bounded by $d_{i-1}$, the intrinsic dimensionality of $\mathcal{H}_{i-1}$. Thus, we have:

$$\text{Number of non-noisy directions at } \mathcal{H}_{i-1} \leq d_{i-1}$$

Consequently, we can rewrite the inequality from Theorem 3.2 as follows:

$$d_i \leq d_{i-1} + N_{i-1},$$

where $N_{i-1}$ represents the number of non-noisy directions in the parameter space that needs to be optimized.

Since $N_{i-1}$ represents the number of parameters to be optimized at block $i$, and by definition $N_{i-1} \geq 0$, we conclude:

$$\max(d_i - d_{i-1}, 0) \leq N_{i-1}.$$

This completes the proof. $\qquad\square$

### B.3 Intuitive Proof of Conjecture 3.1 – Transformer Rank Bound Dynamics

**Conjecture B.1** (**Transformer Rank Bound Dynamics**). *Let $i \in \{1, 2, \ldots, N\}$, and consider the process of fine-tuning. During this process, both the rank of each transformer block $\text{rank}(\mathcal{T}_i)$ and the intrinsic dimension $d_i$ of the manifold $\mathcal{H}_i$ decrease. Let $d_i^0$ denote the initial intrinsic dimension. Then, the following inequality holds:*

$$d_i^0 \leq \text{rank}(\mathcal{T}_i^t),$$

*where $\mathcal{T}_i^t$ represents the transformer block after the $t$-th gradient step. As fine-tuning progresses, this inequality becomes progressively tighter, implying that the gap between the initial intrinsic dimension and the rank of the transformer block reduces over time.*

*Proof.* Here, we outline an intuitive proof of our conjecture. Before fine-tuning, the hidden states explore a large, unconstrained space, leading to a high intrinsic dimension $d_i^0$ of the manifold $\mathcal{N}_i$ and a relatively high rank for the transformer block $\mathcal{T}_i^0$. During fine-tuning, the model becomes specialized for a specific task. It learns to focus on relevant features, causing the hidden states to lie on a lower-dimensional subspace, which reduces the intrinsic dimension $d_i$. Simultaneously, the rank of $\mathcal{T}_i$ decreases as the block's transformation focuses on fewer independent directions, filtering out irrelevant information. As both the intrinsic dimension and rank decrease during fine-tuning, the inequality $d_i^0 \leq \text{rank}(\mathcal{T}_i)$ becomes tighter.

This completes the proof. □

## C    DATASETS STATISTICS

### C.1    GLUE BENCHMARK

We present the statistics for the GLUE (Wang et al., 2019) datasets used in our experiments in Table 6.

Table 6: Summary of the GLUE benchmark datasets.

| Corpus | Task | #Train | #Dev | #Test | #Label | Metrics |
|--------|------|--------|------|-------|--------|---------|
| CoLA | Acceptability | 8.5k | 1k | 1k | 2 | Matthews Corr. |
| SST-2 | Sentiment | 67k | 872 | 1.8k | 2 | Accuracy |
| RTE | NLI | 2.5k | 276 | 3k | 2 | Accuracy |
| MRPC | Paraphrase | 3.7k | 408 | 1.7k | 2 | Accuracy |
| QNLI | QA/NLI | 108k | 5.7k | 5.7k | 2 | Accuracy |
| STS-B | Similarity | 7k | 1.5k | 1.4k | – | Pearson/Spearman Corr. |

### C.2    SQUAD DATASETS

We present the statistics for the SQUAD (Rajpurkar et al., 2016) datasets used in our experiments in Table 7.

Table 7: Statistics of the SQuAD dataset.

|  | # Train | # Validation |
|--|---------|--------------|
| SQuAD v1.1 | 87,599 | 10,570 |
| SQuAD v2.0 | 130,319 | 11,873 |

### C.3    AIROBOROS DATASET

We present the statistics for the Airoboros (Durbin, 2024) dataset used in our experiments in Table 8.

Table 8: Statistics of the Airoboros dataset.

|  | # Train |
|--|---------|
| Airobors | 29,400 |

## C.4 MT-BENCH BENCHMARK

We present the statistics for the MT-BENCH (Zheng et al., 2023b) dataset used in our experiments in Table 9.

Table 9: Statistics of the MT-BENCH dataset.

|  | # Samples |
|---|---|
| MT-BENCH | 80 |

## D EXAMPLES OF RANK PATTERNS

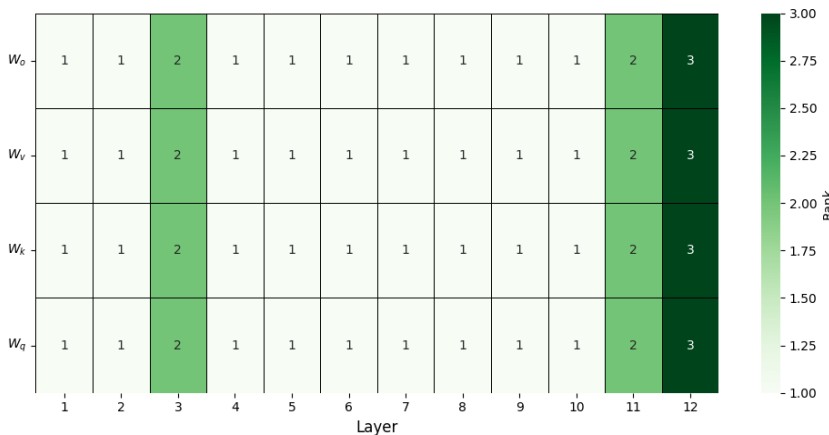

Figure 5: GeLoRA rank pattern for CoLA

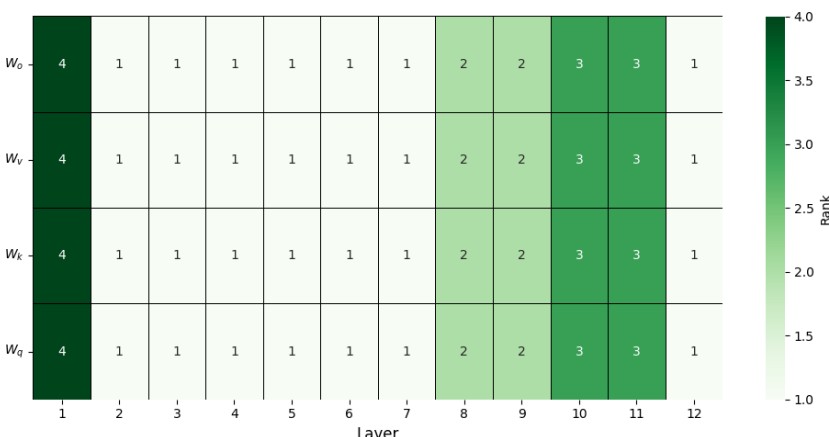

Figure 6: GeLoRA rank pattern for MRPC

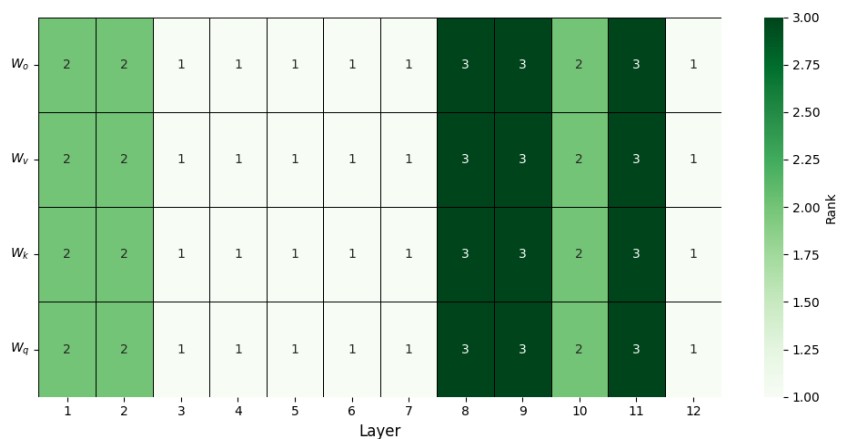

Figure 7: GeLoRA rank pattern for RTE

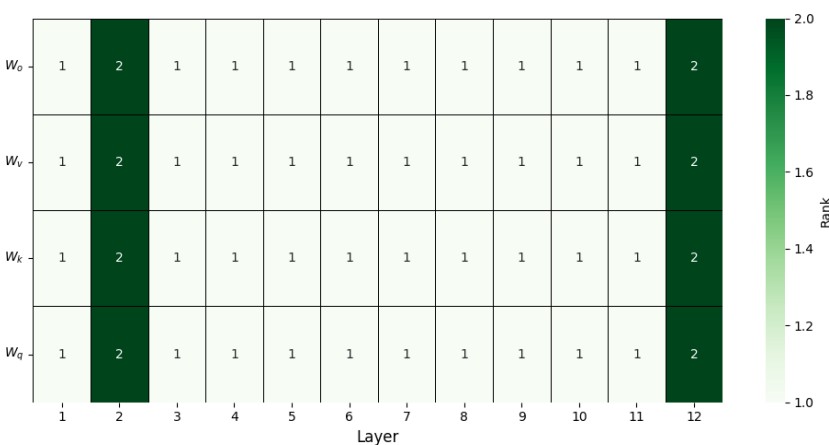

Figure 8: GeLoRA rank pattern for SST-2

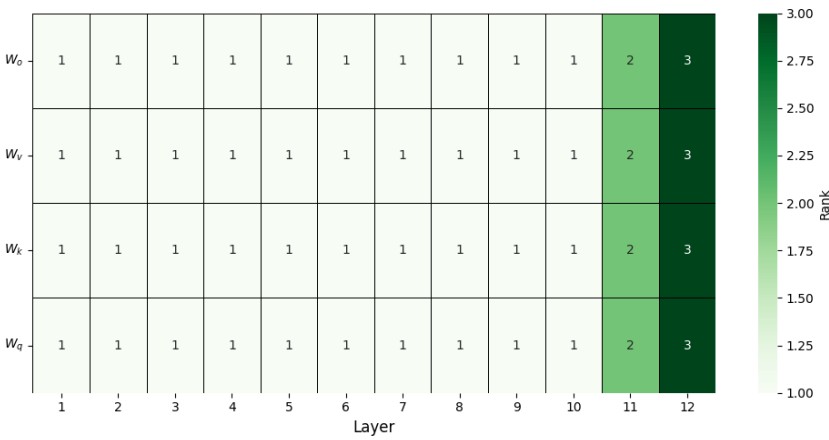

Figure 9: GeLoRA rank pattern for QNLI

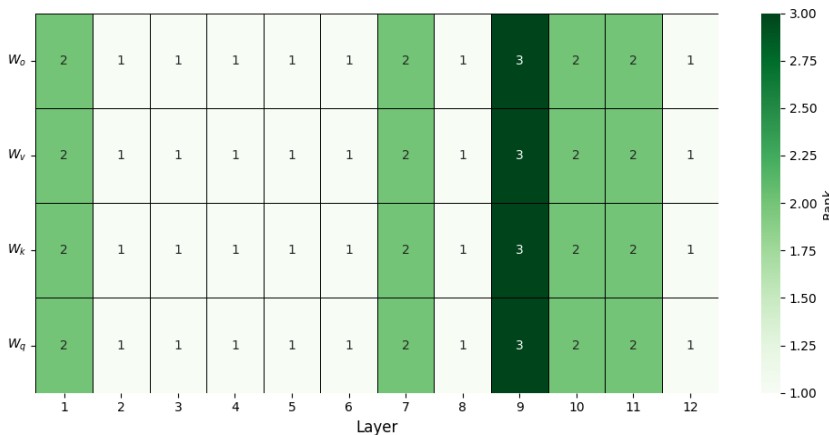

Figure 10: GeLoRA rank pattern for STSB

# E    TRAINING DETAILS

We employ OPTUNA to fine-tune the hyperparameters for the following techniques: LoRA, GeLoRA, BitFit, and Full Finetuning, while using the optimal parameters for SoRA from the original paper. The ranges for hyperparameters include a learning rate between $8e^{-5}$ and $1e^{-3}$, LoRA dropout, warmup ratio, and weight decay between 0 and 0.1, as well as two types of schedulers: linear and cosine.

Hereafter, we summarize the optimal parameters identified across 50 trials, which were used in the fine-tuning process.

Table 10: Hyperparameters for GeLoRA for each task

| Hyperparameter | CoLA | STS-B | MRPC | QNLI | SST-2 | RTE |
|---|---|---|---|---|---|---|
| **Learning Rate** | $8.00e^{-5}$ | $1.69e^{-4}$ | $7.53e^{-4}$ | $1.88e^{-4}$ | $1.61e^{-4}$ | $1.51e^{-4}$ |
| **Weight Decay** | $1.00e^{-1}$ | $9.43e^{-2}$ | $5.48e^{-2}$ | $3.00e^{-2}$ | $3.22e^{-2}$ | $6.78e^{-2}$ |
| **Warmup Ratio** | $6.00e^{-2}$ | $1.65e^{-2}$ | $3.04e^{-2}$ | $5.91e^{-2}$ | $7.63e^{-2}$ | $6.35e^{-2}$ |
| **LoRA Dropout** | $5.00e^{-2}$ | $5.69e^{-2}$ | $1.88e^{-2}$ | $5.36e^{-2}$ | $4.68e^{-2}$ | $7.16e^{-2}$ |
| **Scheduler Type** | Linear | Cosine | Linear | Linear | Cosine | Cosine |

Table 11: Hyperparameters for LoRA for each task

| Hyperparameter | CoLA | STS-B | MRPC | QNLI | SST-2 | RTE |
|---|---|---|---|---|---|---|
| **Learning Rate** | $3.88e^{-4}$ | $9.80e^{-5}$ | $4.14e^{-4}$ | $2.12e^{-4}$ | $1.27e^{-4}$ | $3e^{-4}$ |
| **Weight Decay** | $4.88e^{-2}$ | $3.30e^{-2}$ | $8.94e^{-2}$ | $3.03e^{-4}$ | $3.90e^{-2}$ | $2.96e^{-2}$ |
| **Warmup Ratio** | $9.63e^{-2}$ | $3.99e^{-2}$ | $6.28e^{-2}$ | $7.89e^{-2}$ | $8.33e^{-2}$ | $4.9e^{-2}$ |
| **LoRA Dropout** | $9.85e^{-2}$ | $1.00e^{-1}$ | $5.51e^{-2}$ | $7.19e^{-2}$ | $8.09e^{-3}$ | $5.13e^{-2}$ |
| **Scheduler Type** | Cosine | Linear | Linear | Linear | Linear | Cosine |

Table 12: Hyperparameters for Full Finetuning for each task

| Hyperparameter | CoLA | STS-B | MRPC | QNLI | SST-2 | RTE |
|---|---|---|---|---|---|---|
| **Learning Rate** | $1.12e^{-4}$ | $1.03e^{-4}$ | $6.87e^{-4}$ | $1.03e^{-4}$ | $1.27e^{-4}$ | $9.29e^{-5}$ |
| **Weight Decay** | $5.53e^{-2}$ | $3.21e^{-2}$ | $7.48e^{-2}$ | $5.63e^{-3}$ | $3.90e^{-2}$ | $6.35e^{-2}$ |
| **Warmup Ratio** | $2.34e^{-2}$ | $9.30e^{-2}$ | $7.44e^{-2}$ | $4.76e^{-2}$ | $8.33e^{-2}$ | $3.33e^{-2}$ |
| **Scheduler Type** | Cosine | Cosine | Cosine | Cosine | Linear | Cosine |

Table 13: Hyperparameters for BitFit for each task

| Hyperparameter | CoLA | STS-B | MRPC | QNLI | SST-2 | RTE |
|---|---|---|---|---|---|---|
| **Learning Rate** | $7.94e^{-4}$ | $5.53e^{-4}$ | $8.61e^{-4}$ | $7.91e^{-4}$ | $3.36e^{-4}$ | $1.00e^{-3}$ |
| **Weight Decay** | $2.00e^{-2}$ | $8.89e^{-2}$ | $9.89e^{-2}$ | $4.70e^{-3}$ | $3.16e^{-2}$ | $1.11e^{-2}$ |
| **Warmup Ratio** | $1.00e^{-1}$ | $2.75e^{-2}$ | $8.10e^{-2}$ | $7.07e^{-2}$ | $8.33e^{-2}$ | $6.19e^{-2}$ |
| **Scheduler Type** | Cosine | Linear | Cosine | Linear | Cosine | Linear |

