# OpenReview forum: "GLoRA: Geometric Adaptive Ranks for Efficient LoRA Fine-Tuning"
_ICLR.cc/2025/Conference — Submitted to ICLR 2025_

### Official Review · Reviewer_oj1a · 2024-11-03

**Soundness:** 1
**Presentation:** 1
**Contribution:** 2
**Rating:** 3
**Confidence:** 3

**Summary:**

This paper studies the trade-off between model performance and LoRA rank in the efficient fine-tuning scenario. Specifically, the authors propose GLoRA to adaptively select LoRA ranks by evaluating the lower bound of the optimal LoRA rank using the intrinsic dimension of the hidden representations. The author evaluate the performance of GLoRA on finetuning BERT-like classification models on the GLUE benchmark.

**Strengths:**

1. This paper studies the lower bound of the optimal LoRA fine-tuning rank, which bridge between the geometry of hidden representations and model parameter manifold.
2. This paper evaluate the performance of GLoRA in fine-tuning on GLUE dataset.

**Weaknesses:**

1. The authors did not provide a formal definition or introduction of the proposed GLoRA algorithm.
2. From my understanding, according to Theorem 3.1 and Theorem 3.2, the authors proposed to set the amount of tunable parameters as $\max(d_{i}-d_{i-1},0)$, where $d_i$ is lower bounded by the rank of the Fisher information matrix of the $i$-th transformer block. However, the lower bounding $d_i$ and $d_{i-1}$ is not sufficient to establish an informative estimation of $d_i-d_{i-1}$.
3. The lower bound of the so-called 'optimal rank' in Theorem 3.1 and Theorem 3.2 seems to be loose, as it can takes the value $0$, which is the trivial lower bound.
4. The established lower bound does not reflect how the LoRA rank can be adaptively adjusted based on data representations that might change during the fine-tuning process. The authors did not clarify in what sense the GLoRA ranks are considered "adaptive."
5. The authors did not provide compare their method against AdaLoRA, SaLoRA, and ALoRA in Table 1 and Table 3. The empirical experiments are confined on the DeBERTaV3 architecture and GLUE dataset.
6. Although this paper aims to study the optimal parameter rank through the lens of Riemannian geometry theory, the solidness of the notations and the derivations in this paper is compromised by some typo and undefined notations. Therefore, the paper is not easy to follow.
   1) What is the notation $\mathrm{dim} \mathfrak{J}mf$ in Page 3, line 157?
   2) The subscription $\phi$ might be missed in the FIM in Page 4, lines 181-184.
   3) The Riemannian metric is a (2,0)-tensor which measure the distance between two inputs (tangent vectors). For clarity, the FIM euqation might be $\mathcal{I}(\phi_1,\phi_2)=\mathbb{E}[\partial_{\phi_1}\log \mathbb{P}\cdot \partial_{\phi_2}\log \mathbb{P}^\top]$.
   4) For self-containess and rigority, can the authors add the formal mathematical definition of 'intrinsic dimension' in lines 240-242, and the 'estimated intrinsic dimension' $\mathrm{idim}(\cdot)$ in Theorem 3.1?

**Questions:**

1. How does the Conjecture 3.1 helps estimating the lower bound of $N_{i-1}$ or improving the rank selection process (e.g. which rank to choose) in LoRA during the training process?
2. What is the algorithmic complexity of the rank estimation process in GLoRA?
3. Does GLoRA use a fixed rank across the training process? If not, how does GLoRA adapt the number of rank? What is the rank increasing and decreasing policy? Otherwise, what is the resulting rank pattern of GLoRA?

---

> ### Author Response · Authors · 2024-11-25
> **Response to Weakness 1**
>
> Firstly we would like to thank the reviewer **oj1a** for his in-depth feedback in comments.
>
> **Weakness 1:** The authors did not provide a formal definition or introduction of the proposed GLoRA algorithm.
>
> **Response:**
> We acknowledge that the original version of our paper lacked a detailed description of the main GeLoRA framework and its connection to the underlying theory. This has been addressed in the revised version. To summarize, the GeLoRA framework recognizes that the parameters adjusted during fine-tuning should not be uniformly distributed across all layers or fixed for all tasks. This is because the model may already possess prior knowledge relevant to the task, either from pretraining data or from earlier fine-tuning on related knowledge. Similarly, each layer contains distinct parameters that capture task-specific features.
> Building on our theoretical results, we derive a lower bound for the optimal rank of each transformer layer based on the input and output dimensions of the hidden states. This lower bound serves as the rank configuration for fine-tuning, and the ranks are pre-computed before the training process using intrinsic dimension estimation tools (specifically 2NN Estimator).
>
> The algorithm comprises four main steps:
>
> - **Compute hidden states:** Extract the hidden states of the training data from the model.
> - **Estimate intrinsic dimensions:** Use the 2-NN algorithm, a widely adopted method for analyzing deep learning representations, to estimate the intrinsic dimensions of the hidden states.
> - **Determine rank configuration:** Set the rank of each layer based on the computed intrinsic dimensions and our main Theorem. To ensure no significant information is overlooked, an offset of 1 is added to each rank, as the estimated intrinsic dimensions are lower bounds of the true intrinsic dimension.
> - **Fine-tune with LoRA:** Fine-tune the model using LoRA, using the precomputed rank pattern.
>
> This approach ensures a task-specific and layer-adaptive fine-tuning process, aligning with the theoretical analysis of GeLoRA.  We have also included a **pseudo-code** highlighting these main steps in **Appendix A** of our revised version.

---

> ### Author Response · Authors · 2024-11-25
> **Response to Weakness 2**
>
> **Weakness 2:** From my understanding, according to Theorem 3.1 and Theorem 3.2, the authors proposed to set the amount of tunable parameters as $\max(d_i−d_{i−1},0)$, where $d_i$ is lower bounded by the rank of the Fisher information matrix of the $i$-th transformer block. However, the lower bounding $d_i$ and $d_{i−1}$  is not sufficient to establish an informative estimation of $d_i−d_{i−1}$.
>
> **Response:**
>
> We apologize for any confusion. Let us explain what we meant by both theorems to clarify any confusions.
> - **Theorem 3.1** is introduced to explain the concept of intrinsic dimension, where the rank is bounded by the **intrinsic dimension of the parameter manifold**. However, computing the intrinsic dimension of the parameter manifold, or the rank of the Fisher information for each layer, is not straightforward. Most methods that compute Fisher information do so for the entire neural network, and this approach is computationally expensive and not granular at all. We refer the paper to the **lines 233-245** of our revised document.
> - To address this, we proposed shifting the focus to the data manifold, which is more tractable. In **Theorem 3.2**, we demonstrate that it is possible to derive a lower bound for the transformer rank based on the **intrinsic dimensions of the hidden states**. These hidden states are successive transformations of the training data through the transformer blocks. This approach provides a more efficient and computationally feasible way to estimate the ranks for fine-tuning.
> - To further clarify the notations, in **Theorem 3.1**, idim refers to the **intrinsic dimension of the parameter manifold**. In contrast, $d_i$ in **Theorem 3.2** represents **the intrinsic dimension of the hidden state**, which is the transformation of the data manifold by the first $i−1$ transformer blocks and serves as the input to the $i$-th transformer block. This distinction helps differentiate between the intrinsic dimensions of the parameter manifold and the data data manifold, which can be both used to have lower bounds on the transformer rank, but using the data manifold (which GeLoRA does) is more straightforward.

---

> ### Author Response · Authors · 2024-11-25
> **Response to Weakness 3**
>
> **Weakness 3:** The lower bound of the so-called 'optimal rank' in Theorem 3.1 and Theorem 3.2 seems to be loose, as it can takes the value 0, which is the trivial lower bound.
>
> **Response:**
> - According to our experiments, the ranks take the value of the trivial bound for some layers but not all of them. However, we believe that when the rank is set to $0$, it indicates that the corresponding transformer block is not necessary for fine-tuning for that specific task. This observation is further supported by the work of **Hartford et al. (2024)** in their paper **Spectrum: Targeted Training on Signal to Noise Ratio**, which shows that some layers may not be informative and can be left out during the training process. Their approach computes the signal-to-noise ratio for each block and uses this information to selectively fine-tune layers based on the value they contribute to the model’s performance.
> - Additionally, it is important to note that since we are using estimators for the intrinsic dimension rather than the ground truth, we only have a lower bound for the true intrinsic dimension. To mitigate the potential errors in estimation and avoid neglecting important layers, we add an offset of 1. This ensures that we do not exclude critical layers due to inaccuracies in the intrinsic dimension estimation.
> - We refer the reviewer to the rank patterns of GeLoRA for the GLUE benchmark.
>
> **Reference:**
> - Eric Hartford, Lucas Atkins, Fernando Fernandes Neto, & David Golchinfar. (2024). Spectrum: Targeted Training on Signal to Noise Ratio.

---

> ### Author Response · Authors · 2024-11-25
> **Response to Weakness 4**
>
> **Weakness 4:** The established lower bound does not reflect how the LoRA rank can be adaptively adjusted based on data representations that might change during the fine-tuning process. The authors did not clarify in what sense the GLoRA ranks are considered "adaptive."
>
> **Response:**
>
> - GeLoRA framework recognizes that the parameters adjusted during fine-tuning should not be uniformly distributed across all layers or fixed for all tasks. This is because the model may already possess prior knowledge relevant to the task, either from pretraining data or from earlier fine-tuning on related knowledge. Similarly, each layer contains distinct parameters that capture task-specific features. Hence, by “adaptive” ranks, we mean that the ranks we use for fine-tuning depend on the model, the dataset, and the transformer block, and they are computed prior to fine-tuning. We refer the reviewer to the description of our approach in the revised version, and also the computed rank patterns for the GLUE benchmark in the Appendix.
> - However, we do not change the ranks during the fine-tuning process, as we believe that our lower bound will remain optimal. This is because gradient-based techniques like SGD tend to prefer flatter minima, meaning as the training progresses, the number of directions influencing the optimization process, and hence the ranks of the transformer blocks decreases and approach our lower bound. Consequently, the bound we compute becomes increasingly tight and optimal as training advances.

---

> ### Author Response · Authors · 2024-11-25
> **Response to Weakness 5**
>
> **Weakness 5:** The authors did not provide compare their method against AdaLoRA, SaLoRA, and ALoRA in Table 1 and Table 3. The empirical experiments are confined on the DeBERTaV3 architecture and GLUE dataset.
>
> **Response:**
> - We acknowledge that our experimental section may have been limited. Yet, we chose to use DeBERTaV3 architecture as it remains highly relevant, particularly for natural language understanding (NLU) tasks like those in the GLUE benchmark. DeBERTaV3 was specifically designed to excel in these tasks by using techniques such as disentangled attention providing a good foundation for evaluating our framework, especially that other works (LoRA, SoRA, AdaLoRA) have used the same model for their evaluations.
> - Moreover, we have added additional baselines for more robust benchmarking, as well as expanded the scope to include more tasks. Specifically, we conducted additional experiments on question-answering tasks using the **SQuAD dataset** (both v1.1 and v2), and these results are now included in the updated version of our paper. We will also be adding results for the **QQP** and **MNLI** datasets shortly.
> Furthermore, we are actively working on expanding our evaluation by incorporating an experiment focused on instruction-tuning tasks and decoder-only models. In particular, we are fine-tuning the **Phi-2 model** using the **Airoboros dataset** and evaluating its performance on the **MT-Bench dataset**. In addition to these new tasks, we have broadened our comparison by including more baselines, specifically adapter-based techniques such as **Houlsby** and **Pfeiffer** adapters, as well as **AdaLoRA**.
> - However, for **SaLoRA** and **ALoRA**, we were unable to include them as baselines due to the unavailability of publicly accessible code to reproduce their experiments.

---

> ### Author Response · Authors · 2024-11-25
> **Response to Weakness 6**
>
> **Weakness 6:** Although this paper aims to study the optimal parameter rank through the lens of Riemannian geometry theory, the solidness of the notations and the derivations in this paper is compromised by some typos and undefined notations. Therefore, the paper is not easy to follow.
>
> 1. What is the notation $\dim Imf$ in Page 3, line 157?
> 2. The subscript $\phi$ might be missed in the FIM in Page 4, lines 181–184.
> 3. The Riemannian metric is a $(2,0)$-tensor which measures the distance between two inputs (tangent vectors). For clarity, the FIM equation might be:
>   $$
>   I(\phi_1, \phi_2) = \mathbb{E}\left[\partial_{\phi_1} \log P \cdot \partial_{\phi_2} \log P^\top\right]
>   $$
> 4. For self-containment and rigor, can the authors add the formal mathematical definition of 'intrinsic dimension' in lines 240–242, and the 'estimated intrinsic dimension' $\text{idim}(\cdot)$ in Theorem 3.1?
>
> **Response:**
> 1.  The dimension of the image of $f$, denoted as $\dim Imf$, represents the rank of the linear transformation $f$, which is the dimension of its image. We used this example to illustrate the intuition behind GeLoRA. Specifically, the dimension of the image of the input under the transformation $f$ is equal to the rank of the matrix $A$. This establishes a connection between the parameters, represented by the matrix $A$, and the data space, represented by $X$, and its transformation under the application of $f$, denoted as $f(X)$. This relationship highlights how the structure of the parameters influences the data space
> 2. 3. We fully agree with the definition provided. However, the expression we presented is in matrix form, where $\phi$ represents the vector of all model parameters. We acknowledge that it is also possible to represent this by specifying the actual values of the matrix entries, as suggested by the reviewer. For reference, we adopted a similar notation to that used by **Shunshi Amari** in his book **Information Geometry and its Applications**.
> 4. In our previous version of the manuscript, we provided the following definition of intrinsic dimension: "The intrinsic dimension is defined as the minimum number of parameters required to capture the local variance of the data points effectively." Within the context of the parameter manifold, the intrinsic dimension is determined by thresholding the eigenvalues of the Fisher Information Matrix (FIM) to exclude uninformative directions. The number of significant eigenvalues, which correspond to informative directions, defines the intrinsic dimension of the parameter manifold. Furthermore, in the revised version of the manuscript, we have clarified our approach to estimating the intrinsic dimension of hidden states using the Two Nearest Neighbors (TwoNN) technique. If the reviewer judge it necessary to formulate these definitions in "definition" latex environment, we will gladly do it.

---

> ### Author Response · Authors · 2024-11-25
> **Response to Question 1**
>
> **Question 1:** How does the Conjecture 3.1 helps estimating the lower bound of or improving the rank selection process (e.g. which rank to choose) in LoRA during the training process?
>
> **Response:**
> As we explained above (Weakness 1 response), GeLoRA compute the rank patterns before the fine-tuning process. The main idea behind this conjecture is that these computed ranks will remain optimal throughout the learning process. Indeed, gradient descent techniques such as SGD prefer flatter minima, which means as the training progresses, the number of directions influencing the optimization process, and hence the ranks of the transformers decrease and approach their respective lower bounds. Consequently, the bound we compute becomes increasingly tight and optimal as training advances.

---

> ### Author Response · Authors · 2024-11-25
> **Response to Question 2**
>
> **Question 2:** What is the algorithmic complexity of the rank estimation process in GLoRA?
>
> **Response:**
>
> - The rank estimation is a preprocessing step, as explained earlier. In our experiments, we used the TwoNN estimator for intrinsic dimensions, which we have further detailed in the updated version of our paper.
> - Using this estimator, the computation of the rank has a complexity of O(LN^2), where L is the number of transformer blocks in the model and N is the number of data samples. However, this complexity can be significantly reduced to linear O(LN) by employing persistent homology-based dimension estimation accelerated using GPUs. This ensures that the pre-processing step remains scalable, even for large datasets and models.
>
> **References for persistent homology dimension:**
>
> - Michael G. Rawson. (2022). Linear Run Time of Persistent Homology Computation with GPU Parallelization.
> - Eduard Tulchinskii, Kristian Kuznetsov, Laida Kushnareva, Daniil Cherniavskii, Serguei Barannikov, Irina Piontkovskaya, Sergey Nikolenko, & Evgeny Burnaev. (2023). Intrinsic Dimension Estimation for Robust Detection of AI-Generated Texts).

---

> ### Author Response · Authors · 2024-11-25
> **Response to Question 3**
>
> **Question 3:** Does GLoRA use a fixed rank across the training process? If not, how does GLoRA adapt the number of rank? What is the rank increasing and decreasing policy? Otherwise, what is the resulting rank pattern of GLoRA?
>
> **Response:**
>
> By **“adaptive”** ranks, we mean that the ranks we use for fine-tuning depend on the model, the dataset, and the transformer block. However, we do not change the ranks during the fine-tuning process, as we believe that our lower bound will remain optimal. This is because gradient-based techniques like SGD tend to prefer flatter minima, meaning as the training progresses, the number of directions influencing the optimization process decreases. Consequently, the bound we compute becomes increasingly tight and optimal as training advances.
>
> Thank you for the detailed feedback.

---

> ### Comment · Reviewer_oj1a · 2024-11-27
> **Response to the Authors' Rebuttal**
>
> Dear authors,
>
> I appreciate your time and effort in providing additional clarifications and revisions and detailed feedback. After reading the response from the authors and the revised manuscript, while some of my misunderstanding and concerns have be addressed, I still have the following confusions:
>
> 1. In Figure 2, Section 3.3, does the 2-NN estimation require to traverse pass the whole dataset? If the intrinsic dimension is fitted on a mini-batch or a subset, is the variance of the estimation controllable?
> 2. As shown in Lines 336-337, the low-rank adapters are imposed to each matrix (e.g. q_proj, k_proj, v_proj) instead of the whole self-attention layer. However, as shown in Figure 2, the LoRA rank lower bound $r_i=\max(d_i-d_{i-1},0)$ is estimated based on the intrinsic dimension of the input and output of the transformer block, rather than each matrix. This seems to be a discrepancy between implementation and theory, as adding $r_i$-rank LoRA to each matrix is not equivalent to enlarging the rank of the non-linear transformer layer by $r_i$.
> 3. The authors is recommended to provide some empirical validation in supporting the rationale of using the 2-NN estimator, i.e., showing that the cumulative distribution law in line 323 approximatly holds in real-world datasets (e.g., GLUE). Some quantitative analysis on how well the 2-NN estimator fits the intrinsitc dimension on NLP datasets would be helpful in justifying the effectiveness of GeLoRA.
> 4. In Table 1 and Table 3, the performance of GeLoRA is not competitive across different datasets, even when comparing with LoRA variants with rank $=1$. This weaken the necessity and efficacy of using GeLoRA.
>
> Overall, the current manuscript has a large room for improvement, as the current version still contains some missing references (Section 3.4),  text overlapping (line 475-476), unexplained mathematical notations (e.g., local variance, the notation of image). Therefore, I have to keep my rating.
>
> However, when comparing the rebuttal revisions with the original one, I still witness a great improvement. In general, exploiting the connection between the data and parameter geometry is an elegant idea. I believe you have great potential in improving GeLoRA's performance, and improving the manuscript towards the acceptance bar of top-tier machine learning conferences.
>
> Best,
>
> Reviewer oj1a

---

> > ### Author Response · Authors · 2024-12-01
> > **Response to the reviewer's feedback**
> >
> > Firstly, we would like to thank the reviewer for his feedback and comments.
> >
> > **Answer to Question 1:**
> > - In our revised manuscript, we have included additional appendix sections that provide a detailed explanation of intrinsic dimensionality, its computation, and a hands-on example to illustrate the TwoNN approach.
> >
> > - It is worth noting that computing intrinsic dimensionality for the entire dataset is not always necessary and, in some cases, may even be inadvisable. Performing this computation on the full dataset can potentially overfit noise. For instance, the authors of the "Geometry of Hidden Representations" paper conducted their analysis using only N/8 and N/2 samples rather than the entire dataset.
> >
> > - Furthermore, in the original work by Facco et al. (Facco, E., d’Errico, M., Rodriguez, A., & Laio, A. (2017). Estimating the intrinsic dimension of datasets by a minimal neighborhood information. Scientific Reports, 7(1)), the authors performed a multiscale analysis, identifying the relevant dimensions as a function of the subset (block) size showing that the intrinsic dimension usually plateau.
> >
> > **Answer to Question 2:**
> >
> > We completely agree that a transformer block typically consists of multiple matrices, as outlined in the Definitions section of our revised paper, and the proof section. However, the required number of trainable parameters is determined for the entire transformer block, and since it is not feasible to individually target each matrix, we allocate the total number of parameters to optimize for the entire block to each individual matrix. This approach ensures that all necessary parameters across the block are effectively optimized.
> >
> > **Answer to Question 3:**
> >
> > - We used the Two Nearest Neighbors (TwoNN) estimator, following established literature. For example, in the "Geometry of Hidden Representations" paper, the authors computed the intrinsic dimension of transformer models across various datasets, including proteins, natural language processing (NLP) tasks, and images. The primary condition that must be verified when using this method is the assumption of constant density, which, according to the paper, holds for up to four neighbors, thus we can use the Two Nearest Neighbors.
> >
> > - Another paper that uses the Two Nearest Neighbors along with other intrinsic dimension estimor for natural language is the following:
> > Eduard Tulchinskii, Kristian Kuznetsov, Laida Kushnareva, Daniil Cherniavskii, Serguei Barannikov, Irina Piontkovskaya, Sergey Nikolenko, & Evgeny Burnaev. (2023). Intrinsic Dimension Estimation for Robust Detection of AI-Generated Texts.
> >
> > **Answer to Question 4:**
> >
> > Although large language models excel at practical tasks, they lack a strong theoretical foundation. Their abilities to generalize, represent knowledge, and perform complex reasoning remain poorly understood due to opaque training dynamics in high-dimensional parameter spaces. This opacity makes it challenging to explain their behavior, ensure robustness, and systematically improve their design. In our GeLoRA work, we aim to bridge this gap by providing a solid theoretical framework that elucidates the inner working mechanism of LLMs and fine-tuning techniques. We believe that understanding the relationship between the data manifold and training dynamics in parameter space is essential for effective algorithm optimization.
> >
> > While GeLoRA may slightly underperform on specific tasks, it consistently delivers strong results across diverse benchmarks, achieving higher average scores than other fine-tuning methods. Moreover, it accomplishes this while utilizing significantly fewer parameters, making it a more efficient and scalable solution.
> >
> > We also included the remaining remarks in the new updated version of our manuscript.
> >
> > Thank you for your time and consideration.
> >
> > Best,

---

### Official Review · Reviewer_pwGH · 2024-11-04

**Soundness:** 4
**Presentation:** 2
**Contribution:** 3
**Rating:** 6
**Confidence:** 4

**Summary:**

The paper introduces a novel framework called Geometric Low-Rank Adaptation (GLoRA) for the efficient fine-tuning of large language models (LLMs). The key innovation of GLoRA is in its approach to Low-Rank Adaptation (LoRA), a method that updates only a subset of model weights to reduce computational costs. GLoRA computes the intrinsic dimensionality of hidden state representations to adaptively select LoRA ranks, providing a theoretical basis for optimizing the trade-off between model performance and efficiency. The framework dynamically adjusts the rank for each layer based on the intrinsic dimensionality of its input and output representations, recognizing the varying impact of model parameters on fine-tuning. Empirical results on multiple tasks show that GLoRA outperforms recent baselines within the same parameter budget.

**Strengths:**

- The paper provides a solid theoretical framework that connects the intrinsic dimensionalities of data representation manifolds with the ranks of weight updates in transformer blocks.
- The paper backs up its claims with extensive empirical validation across multiple tasks, demonstrating that GLoRA consistently outperforms existing baselines while maintaining the same parameter budget.

**Weaknesses:**

- Although the theoretical analysis part (section 3) is very solid, it lacks some takeaways to clearly describe the main body of the theoretical analysis.
- There is a lack of an overall summary description of the GLoRA method. An algorithm description or pseudo code can be added.
- The experiment is a bit thin and lacks some supplementary experiments, such as studying the application position of GLoRA in the transformer model.

**Questions:**

- Why is there a lack of testing for MNLI and QQP tasks for the GLUE benchmark?
- In section 4.1, the authors mentioned that they wanted to compare Adalora. Why are there no experiments on Adalora in the following experiments?

---

> ### Author Response · Authors · 2024-11-25
> **Response to Weakness 1 and 2**
>
> Firstly we would like to thank the reviewer **pwGH** for his in-depth feedback in comments.
>
> - **Weakness 1:** Although the theoretical analysis part (section 3) is very solid, it lacks some takeaways to clearly describe the main body of the theoretical analysis.
> - **Weakness 2:** There is a lack of an overall summary description of the GLoRA method. An algorithm description or pseudo code can be added.
>
> **Response:**
>
> We acknowledge that the original version of our paper lacked a detailed description of the main GeLoRA framework and its connection to the underlying theory. This has been addressed in the revised version. To summarize, the GeLoRA framework recognizes that the parameters adjusted during fine-tuning should not be uniformly distributed across all layers or fixed for all tasks. This is because the model may already possess prior knowledge relevant to the task, either from pretraining data or from earlier fine-tuning on related knowledge. Similarly, each layer contains distinct parameters that capture task-specific features.
> Building on our theoretical results, we derive a lower bound for the optimal rank of each transformer layer based on the input and output dimensions of the hidden states. This lower bound serves as the rank configuration for fine-tuning, and the ranks are pre-computed before the training process using intrinsic dimension estimation tools (specifically 2NN Estimator).
>
> The algorithm comprises four main steps:
> - **Compute hidden states:** Extract the hidden states of the training data from the model.
> - **Estimate intrinsic dimensions:** Use the 2-NN algorithm, a widely adopted method for analyzing deep learning representations, to estimate the intrinsic dimensions of the hidden states.
> - **Determine rank configuration:** Set the rank of each layer based on the computed intrinsic dimensions and our main Theorem. To ensure no significant information is overlooked, an offset of 1 is added to each rank, as the estimated intrinsic dimensions are lower bounds of the true intrinsic dimension.
> - **Fine-tune with LoRA:** Fine-tune the model using LoRA, using the precomputed rank pattern.
>
> This approach ensures a task-specific and layer-adaptive fine-tuning process, aligning with the theoretical analysis of GeLoRA.  We have also included a **pseudo-code** highlighting these main steps in the Appendix.

---

> ### Author Response · Authors · 2024-11-25
> **Response to Weakness 3**
>
> **Weakness 3:** The experiment is a bit thin and lacks some supplementary experiments, such as studying the application position of GLoRA in the transformer model.
>
> **Response:**
>
> We acknowledge that our experimental section may have been limited. To address this, we have added additional baselines for more robust benchmarking, as well as expanded the scope to include more tasks. Specifically, we conducted additional experiments on question-answering tasks using the **SQuAD dataset** (both v1.1 and v2), and these results are now included in the updated version of our paper. We will also be adding results for the **QQP** and **MNLI** datasets shortly.
> Furthermore, we are actively working on expanding our evaluation by incorporating an experiment focused on instruction-tuning tasks and decoder-only models. In particular, we are fine-tuning the **Phi-2 model** using the **Airoboros** dataset and evaluating its performance on the MT-Bench dataset. In addition to these new tasks, we have broadened our comparison by including more baselines, specifically adapter-based techniques such as **Houlsby** and **Pfeiffer** adapters, as well as **AdaLoRA**. However, for **SaLoRA** and **ALoRA**, we were unable to include them as baselines due to the unavailability of publicly accessible code to reproduce their experiments.

---

> ### Author Response · Authors · 2024-11-25
> **Response to Question 1**
>
> **Question 1:** Why is there a lack of testing for MNLI and QQP tasks for the GLUE benchmark?
>
> **Response:**
>
> These datasets will be included in the updated version of the paper shortly. That being said, we have conducted further experiments on question-answering tasks (SQUAD v1.1 and SQUAD v2), which are now incorporated into the revised version. We also plan to add another experiment to evaluate the technique on instruction-tuning tasks, using the Phi-2 model as outlined above.

---

> ### Author Response · Authors · 2024-11-25
> **Response to Question 2**
>
> **Question 2:** In section 4.1, the authors mentioned that they wanted to compare Adalora. Why are there no experiments on Adalora in the following experiments?
>
> **Response:**
>
> We apologize for the oversight. We inadvertently forgot to include the results in the GLUE benchmark table, but they have now been added in the updated version of the paper.
>
> Thank you for the detailed feedback.

---

> ### Comment · Reviewer_pwGH · 2024-11-28
> **Response to the Authors' Rebuttal**
>
> Dear authors,
>
> Thank you for your detailed feedback! Your response basically solves all my questions and concerns.
>
> Best,
>
> Reviewer pwGH

---

> > ### Author Response · Authors · 2024-12-01
> > **Response to the reviewer's feedback**
> >
> > Thank you for your time and feedback. We have carefully updated our manuscript to address all the reviewers' comments.
> >
> > Best regards,

---

### Official Review · Reviewer_1ktr · 2024-11-04

**Soundness:** 2
**Presentation:** 1
**Contribution:** 2
**Rating:** 5
**Confidence:** 3

**Summary:**

This paper introduces GLoRA, a variant of LoRA that leverages the geometric properties of both the dataset and model parameters to adaptively adjust the rank.

**Strengths:**

The paper introduces a theoretical framework that justifies the intuition that the intrinsic dimension is smaller than the embedding dimension and tends to decrease during fine-tuning.

Additionally, the paper proposes an adaptive rank approach to enhance the performance of LoRA across a variety of natural language understanding tasks.

**Weaknesses:**

There are two major weaknesses in this paper:

1. There is a significant gap between the theoretical analysis and the experiments.
    - The connection between the theory (or the intuition behind it) and the method itself is not well-articulated. Specifically, how the theoretical insights are applied in the method is unclear, and there is no discussion on how to utilize existing intrinsic dimension estimation tools effectively during training.

2. The experimental results require further explanation.
    - For instance, in Table 2 on page 8, the "Mean Rank" is given as either an integer (1 or 2), rather than a real number.
    - The authors evaluated their method on only 6 of the 8 tasks from the original LoRA paper, leaving some tasks untested.
    - Additionally, while the proposed method performs significantly better than LoRA with ranks of 1 or 2, the mean rank across all datasets remains either 1 or 2, which raises questions about the method’s effectiveness and the representativeness of the reported results.

**Questions:**

As introduced in the identified weaknesses, here are specific questions:

Q1: How is the theoretical analysis connected to the model? How is the adaptive dimension calculated, and how does the rank change during fine-tuning?

Q2: Why are the Mean Ranks reported as integers?

Q3: Why were the remaining two datasets from the original LoRA paper not tested?

Q4: Why does the proposed method outperform LoRA with ranks of 1 or 2, even though the Mean Rank is consistently 1 or 2 across datasets?

---

> ### Author Response · Authors · 2024-11-25
> **Response to Weakness 1**
>
> Firstly we would like to thank the reviewer **1ktr** for his in-depth feedback in comments.
>
> **Weakness 1:** There is a significant gap between the theoretical analysis and the experiments.
> The connection between the theory (or the intuition behind it) and the method itself is not well-articulated. Specifically, how the theoretical insights are applied in the method is unclear, and there is no discussion on how to utilize existing intrinsic dimension estimation tools effectively during training.
>
> **Response:**
>
> We acknowledge that the original version of our paper lacked a detailed description of the main GeLoRA framework and its connection to the underlying theory. This has been addressed in the revised version. To summarize, the GeLoRA framework recognizes that the parameters adjusted during fine-tuning should not be uniformly distributed across all layers or fixed for all tasks. This is because the model may already possess prior knowledge relevant to the task, either from pretraining data or from earlier fine-tuning on related knowledge. Similarly, each layer contains distinct parameters that capture task-specific features.
> Building on our theoretical results, we derive a lower bound for the optimal rank of each transformer layer based on the input and output dimensions of the hidden states. This lower bound serves as the rank configuration for fine-tuning, and the ranks are pre-computed before the training process using intrinsic dimension estimation tools (specifically 2NN Estimator).
>
> The algorithm comprises four main steps:
> - **Compute hidden states:** Extract the hidden states of the training data from the model.
> - **Estimate intrinsic dimensions:** Use the 2-NN algorithm, a widely adopted method for analyzing deep learning representations, to estimate the intrinsic dimensions of the hidden states.
> - **Determine rank configuration:** Set the rank of each layer based on the computed intrinsic dimensions and our main Theorem. To ensure no significant information is overlooked, an offset of 1 is added to each rank, as the estimated intrinsic dimensions are lower bounds of the true intrinsic dimension.
> - **Fine-tune with LoRA:** Fine-tune the model using LoRA, using the precomputed rank pattern.
>
> This approach ensures a task-specific and layer-adaptive fine-tuning process, aligning with the theoretical analysis of GeLoRA.

---

> ### Author Response · Authors · 2024-11-25
> **Response to Weakness 2**
>
> **Weakness 2:** The experimental results require further explanation.
> For instance, in Table 2 on page 8, the "Mean Rank" is given as either an integer (1 or 2), rather than a real number.
> The authors evaluated their method on only 6 of the 8 tasks from the original LoRA paper, leaving some tasks untested.
> Additionally, while the proposed method performs significantly better than LoRA with ranks of 1 or 2, the mean rank across all datasets remains either 1 or 2, which raises questions about the method’s effectiveness and the representativeness of the reported results.
>
> **Response:**
> - We acknowledge that our Table 2 may have caused some confusion. The rank reported was the **rounded mean rank**, which we computed for use during fine-tuning with LoRA and its variants to ensure fair comparisons. To clarify, we have modified the table to display both the actual mean ranks of GeLoRA and their rounded versions. We have also added figures illustrating the GeLoRA rank patterns for the GLUE benchmark in the Appendix of our revised version.
> - We conducted additional experiments on question-answering tasks using the **SQuAD dataset** (both v1.1 and v2 versions) and have included the results in the updated version of our paper, and we will be adding results for both **QQP** and **MNLI** datasets shortly.
> - Furthermore, we are actively working on expanding our evaluation by incorporating an additional experiment focused on instruction-tuning tasks and decoder-only models. Specifically, we are fine-tuning the **Phi-2 model** using the **Airoboros dataset** and evaluating its performance on the **MT-Bench dataset**.
> - We have also expanded our comparison by including additional baselines, specifically adapter-based techniques such as **Houlsby** and **Pfeiffer** adapters, as well as **AdaLoRA**. However, for **SaLoRA** and **ALoRA**, we were unable to include them as baselines due to the unavailability of publicly accessible code to reproduce their experiments.
> - We believe the confusion regarding the last point arises from Table 2. The rank patterns in GeLoRA are not uniform; they contain varying values, primarily 1, 2, 3 and 4, for the GLUE benchmark. This non-uniformity reflects the layer-specific adjustments that GeLoRA makes based on the intrinsic dimensions, leading to better performance compared to LoRA with uniform rank. We refer the reviewer to the Appendix of our revised version which illustrates the ranks patterns of GeLoRA.

---

> ### Author Response · Authors · 2024-11-25
> **Response to Question 1**
>
> **Question 1:** How is the theoretical analysis connected to the model? How is the adaptive dimension calculated, and how does the rank change during fine-tuning?
>
> **Response:**
> - We refer the reviewer to the explanation provided above **(Weakness 1)** . To avoid any potential confusion we may have caused, by **"adaptive"**  we mean that the ranks we use are determined based on the model, layer, and dataset. These adaptive ranks are computed prior to the training process and do not change during fine-tuning.
> - Specifically, for each transformer block, we compute the ranks before starting the training process based on the model and dataset. These precomputed ranks are derived from the intrinsic dimensions of the data manifold representation at each transformer block, following our theoretical framework. The ranks are dataset and transformer block-specific, as they depend on the characteristics of the training data, and its image by the transformer blocks.
> - Additionally, we conjecture that the lower bound we compute for the ranks remains optimal during fine-tuning. This is because gradient-based techniques like SGD tend to prefer flatter minima, meaning as the training progresses, the number of directions influencing the optimization process decreases, hence the real ranks will decrease during the fine-tuning process and approaches more our lower bound. Consequently, the bound we compute becomes increasingly tight and optimal as training advances.

---

> ### Author Response · Authors · 2024-11-25
> **Response to Question 2**
>
> **Question 2:** Why are the Mean Ranks reported as integers?
>
> **Response:**
>
> We apologize for any confusion. As mentioned previously, the ranks presented in Table 2 represent the **rounded mean ranks**, which we determined to use for fair comparisons with vanilla LoRA and its other variants. We hope this clarification resolves any misunderstandings.

---

> ### Author Response · Authors · 2024-11-25
> **Response to Question 3**
>
> **Question 3:** Why were the remaining two datasets from the original LoRA paper not tested?
>
> **Response:**
>
> These datasets will be included in the updated version of the paper shortly. Additionally, we have conducted further experiments on question-answering tasks, which are now incorporated into the revised version. We also plan to add another experiment to evaluate the technique on instruction-tuning tasks, using the Phi-2 model as outlined above. We refer the reviewer to our previous response for Weakness 2.

---

> ### Author Response · Authors · 2024-11-25
> **Response to Question 4**
>
> **Question 4:** Why does the proposed method outperform LoRA with ranks of 1 or 2, even though the Mean Rank is consistently 1 or 2 across datasets?
>
> **Response:**
>
> - The proposed method outperforms LoRA with ranks of 1 or 2, despite the Mean Rank being consistently 1 or 2 across datasets, because the mean rank here refers to the **rounded mean rank**. In reality, the rank patterns in GeLoRA are not uniform. The ranks are dynamically determined based on the intrinsic dimensions of the data manifold at each transformer block, ensuring that each layer receives an appropriate rank tailored to the specific task and dataset. For the GLUE benchmark, for example, some layers may have a rank of 1, while others have a rank of 2 or 3 **(See Appendix for rank patterns example).** This adaptiveness enables the proposed method to fine-tune the model more efficiently by assigning a rank based on each layer's specific role in capturing task-relevant information.
> - In contrast, LoRA with fixed ranks of 1 or 2 applies the same rank uniformly across all layers, which does not fully leverage the task-specific and layer-specific characteristics of the model. This difference in how ranks are assigned and adapted during fine-tuning results in superior performance with the proposed method, even when the rounded mean rank values are similar across datasets.
>
> Thank you for the detailed feedback.

---

> ### Comment · Reviewer_1ktr · 2024-11-27
> **Revised score with concerns**
>
> I appreciate the dedicated work that the author invested in the rebuttal. Most of my concerns have been addressed, and I have raised my score from 3 to 5.
>
> However, it should be noted that a substantial amount of new information has been provided in the rebuttal. For example:
>
> 1. Detailed discussion about how the algorithm is designed in relation to the proposed theoretical analysis.
> 2. The rank information, along with the rank distribution pattern, is included in the appendix.
>
> These new results are critical for supporting the major claims of this paper. Particularly, the first point should be clearly integrated into the main text. I sincerely suggest that the author polish the paper, incorporate these changes effectively, and consider submitting it to a subsequent venue.

---

> > ### Author Response · Authors · 2024-12-01
> >
> > Thank you for your thoughtful feedback and for taking the time to review our work. We appreciate your recognition of our efforts in the rebuttal process. We have further remain refined our manuscript and updated it to include your feedback and other reviewers feedback.
> >
> > Best regards,

---

### Official Review · Reviewer_qTNK · 2024-11-05

**Soundness:** 3
**Presentation:** 2
**Contribution:** 2
**Rating:** 5
**Confidence:** 4

**Summary:**

This paper proposes GLoRA, a geometric-adaptive LoRA method that is able to adjust each LoRA component's rank throughout the training process. By presenting a series of theoretical analysis, the authors establish the connection between the rank of transformer blocks and the intrinsic dimension. The experiments show that GLoRA is able to achieve better generalization performance with less computational cost.

**Strengths:**

1. This submission presents sufficient rigorous theoretical justification. The definitions, theorems, and corollaries are somewhat stated clearly in Section 3, which solidly supports the algorithm.
2. The experimental results shown so far (Table 1) exhibit huge potential of GLoRA to outperform other LoRA baselines (LoRA, SLoRA) in the scenarios where the computational resources are the bottleneck.

**Weaknesses:**

1. **Missing the main algorithm.** I don't know if it's just me missing this important part, but in the paper, I haven't found any description about the main algorithm of GLoRA. After the theory (Section 3), it directly jumps into the experimental results. If the authors do have this (but buries it in somewhere else), I highly suggest to present it as a separate chapter. It would be also great if they can reiterate the main algorithm during the rebuttal.
2. **Limited scope of empirical verification.** The experiments are conducted only on one LM architecture (DeBERTaV3, proposed in 2021), which is a bit outdated and of limited scope. Especially now the decoder-only LLMs are considered the strongest and are widely used. It would be good to include the experimental results on other LLM architectures.
3. **Not enough baselines.** In the related work, the authors mention a few other baseline adaptive LoRAs (including AdaLoRA, ALoRA, SaLoRA). I wonder why the comparison is only conducted on the naive LoRA and SLoRA, where only one adaptive LoRA baseline is used?
4. **Gap between the theory and the algorithm.** If I understand it correctly, the number of the parameters (rank(T)) needed for a Transformer block to represent a data distribution is not equivalent to the rank of LoRA. How do the authors bridge the gap between this discrepancy during training, as there is no concrete illustration of the algorithm?

**Questions:**

1. Estimating the intrinsic dimension of transformer blocks should be non-trivial and time-consuming. How does GLoRA achieve this with even faster training speed compared to naive LoRA? Is it because of its lower rank?
2. See weekness 4.
3. I understand the use of the AutoML tool for hyper-parameter tuning. I want to know if there are additional hyperparameters introduced by GLoRA, and what is the sensitivity of them if there is any?
4. Missing critical citation of intrinsic dimension: Li, Chunyuan, et al. "Measuring the intrinsic dimension of objective landscapes." arXiv preprint arXiv:1804.08838 (2018)

Most of my concerns and my questions stem from the lack of the main algorithm. I would happily increase my score once it is resolved.

---

> ### Author Response · Authors · 2024-11-25
> **Response to Weakness 1**
>
> Firstly we would like to thank the reviewer **qTNK** for his in-depth feedback in comments.
>
> **Weakness 1:** Missing the main algorithm. I don't know if it's just me missing this important part, but in the paper, I haven't found any description about the main algorithm of GLoRA. After the theory (Section 3), it directly jumps into the experimental results. If the authors do have this (but buries it in somewhere else), I highly suggest to present it as a separate chapter. It would be also great if they can reiterate the main algorithm during the rebuttal.
>
>
> **Response:**
>
>
> We acknowledge that the original version of our paper lacked a detailed description of the main GeLoRA framework and its connection to the underlying theory. This has been addressed in the revised version. To summarize, the GeLoRA framework recognizes that the parameters adjusted during fine-tuning should not be uniformly distributed across all layers or fixed for all tasks. This is because the model may already possess prior knowledge relevant to the task, either from pretraining data or from earlier fine-tuning on related knowledge. Similarly, each layer contains distinct parameters that capture task-specific features.
> Building on our theoretical results, we derive a lower bound for the optimal rank of each transformer layer based on the input and output dimensions of the hidden states. This lower bound serves as the rank configuration for fine-tuning.
>
> The algorithm comprises four main steps:
>
> - **Compute hidden states:** Extract the hidden states of the training data from the model.
> - **Estimate intrinsic dimensions:** Use the 2-NN algorithm, a widely adopted method for analyzing deep learning representations, to estimate the intrinsic dimensions of the hidden states.
> - **Determine rank configuration:** Set the rank of each layer based on the computed intrinsic dimensions and our main Theorem. To ensure no significant information is overlooked, an offset of 1 is added to each rank, as the estimated intrinsic dimensions are lower bounds of the true intrinsic dimension.
> - **Fine-tune with LoRA:** Fine-tune the model using LoRA, using the precomputed rank pattern.
>
> This approach ensures a task-specific and layer-adaptive fine-tuning process, aligning with the theoretical analysis of GeLoRA.

---

> > ### Author Response · Authors · 2024-11-25
> > **Response to Question 4**
> >
> > **Question 4:** Missing critical citation of intrinsic dimension: Li, Chunyuan, et al. "Measuring the intrinsic dimension of objective landscapes." arXiv preprint arXiv:1804.08838 (2018)
> >
> > **Response:**
> >
> > Thank you for the remark, we have added it.
> > Thank you for the detailed feedback.

---

> ### Author Response · Authors · 2024-11-25
> **Response to Weakness 2**
>
> **Weakness 2:** Limited scope of empirical verification. The experiments are conducted only on one LM architecture (DeBERTaV3, proposed in 2021), which is a bit outdated and of limited scope. Especially now the decoder-only LLMs are considered the strongest and are widely used. It would be good to include the experimental results on other LLM architectures.
>
> **Response:**
> - We acknowledge that our experiments were conducted solely on the DeBERTaV3 architecture. While this model was introduced in 2021, it remains highly competitive and relevant, particularly for natural language understanding (NLU) tasks like those in the GLUE benchmark. DeBERTaV3 was specifically designed to excel in these tasks by using techniques such as disentangled attention providing a good foundation for evaluating our framework, especially that other works (LoRA, SoRA, AdaLoRA) have used the same model for their evaluations.
> - Similarly to these works, we conducted additional experiments on question-answering tasks using the **SQuAD dataset** (v1.1 and v2.0) and have included the results in the updated version of our paper.
> - Furthermore, we are actively working on expanding our evaluation by incorporating an additional experiment focused on instruction-tuning tasks and decoder-only models. Specifically, we are fine-tuning the **Phi-2 model** using the **Airoboros dataset** and evaluating its performance on the **MT-Bench dataset**.

---

> ### Author Response · Authors · 2024-11-25
> **Response to Weakness 3**
>
> **Weakness 3:** Not enough baselines. In the related work, the authors mention a few other baseline adaptive LoRAs (including AdaLoRA, ALoRA, SaLoRA). I wonder why the comparison is only conducted on the naive LoRA and SLoRA, where only one adaptive LoRA baseline is used?
>
> **Response:**
>
> We have expanded our comparison by including additional baselines, specifically adapter-based techniques such as **Houlsby** and **Pfeiffer** adapters, as well as **AdaLoRA**. However, for **SaLoRA** and **ALoRA**, we were unable to include them as baselines due to the unavailability of publicly accessible code to reproduce their experiments.

---

> ### Author Response · Authors · 2024-11-25
> **Response to Weakness 4**
>
> **Weakness 4:** Gap between the theory and the algorithm. If I understand it correctly, the number of the parameters (rank(T)) needed for a Transformer block to represent a data distribution is not equivalent to the rank of LoRA. How do the authors bridge the gap between this discrepancy during training, as there is no concrete illustration of the algorithm?
>
> **Response:**
>
> Exactly, the rank of the transformer block represents the number of parameters required for fine-tuning the entire block, which is composed of multiple sub-layers (e.g., value, query, key, and output layers). Since it is challenging to determine the specific contribution of each sub-layer's parameters, we set the rank of each sub-layer equal to the rank of the entire transformer block. This ensures that all essential parameters across the block are included in the fine-tuning process. Our updated version outlines how we set the LoRA ranks.

---

> ### Author Response · Authors · 2024-11-25
> **Response to Question 1**
>
> **Question 1:** Estimating the intrinsic dimension of transformer blocks should be non-trivial and time-consuming. How does GLoRA achieve this with even faster training speed compared to naive LoRA? Is it because of its lower rank?
>
> **Response:**
> - The computation of the intrinsic dimension of the hidden states is a pre-processing step and, therefore, does not affect the training time, as the ranks are precomputed prior to training.
> - Our updated paper version gives an overview of the 2-NN estimator, and outlines the main GeLoRA algorithm.
> - The complexity of the 2-NN algorithm for estimating intrinsic dimensions is O(LN^2), where L is the number of transformer blocks and N is the number of data samples. However, this complexity can be significantly reduced to linear O(LN) by employing persistent homology-based dimension estimation accelerated using GPUs. This ensures that the pre-processing step remains scalable, even for large datasets and models.
>
> **References for persistent homology dimension:**
> - Michael G. Rawson. (2022). Linear Run Time of Persistent Homology Computation with GPU Parallelization.
> -  Eduard Tulchinskii, Kristian Kuznetsov, Laida Kushnareva, Daniil Cherniavskii, Serguei Barannikov, Irina Piontkovskaya, Sergey Nikolenko, & Evgeny Burnaev. (2023). Intrinsic Dimension Estimation for Robust Detection of AI-Generated Texts).

---

> ### Author Response · Authors · 2024-11-25
> **Response to Question 2**
>
> **Question 2:** See Weakness 4.
>
> **Response:**
>
> We refer the reviewer to our response to Weakness 4 above.

---

> ### Author Response · Authors · 2024-11-25
> **Response to Question 3**
>
> **Question 3:**  I understand the use of the AutoML tool for hyper-parameter tuning. I want to know if there are additional hyperparameters introduced by GLoRA, and what is the sensitivity of them if there is any?
>
> **Response:**
>
> GeLoRA relies solely on the estimation of intrinsic dimensions and does not introduce any additional hyperparameters. Furthermore, when using standard intrinsic dimension estimation techniques commonly employed in deep learning, such as MLE, 2-NN, or Persistent Homology, we observe that they produce approximately similar intrinsic dimension patterns, thus similar lower bounds.

---

> ### Author Response · Authors · 2024-12-01
> **Final Feedback on Rebuttal Updates**
>
> Dear reviewer,
>
> We would like to thank you for your time and thoughtful feedback. We understand your concerns and have worked diligently to address them in our updates. However, as the rebuttal process is nearing its conclusion, we kindly request your updated feedback and confirmation on whether we have sufficiently addressed your comments and concerns.
>
> We greatly value your input and look forward to hearing from you.
>
> Best regards,

---

### Comment · Reviewer_oj1a · 2024-11-26
**Could you provide a summary of rebuttal revision?**

Dear authors,

I have read through the comments of all other reviewers and also your response. I aware that you have dedicate many time and effort in revising the manuscript. Could you mind providing a 'summary of rebuttal revision' section, to highlight the revisions you made, and the purpose of these revisions (e.g. you made revision XXX to address the concerns from Reviewer XXX). I believe that can help you clarify your response and faciliate further discussions.

Best,

Reviewer oj1a

---

> ### Author Response · Authors · 2024-11-27
> **Summary of rebuttal revision (1/2)**
>
> Firstly, we would like to thank all reviewers for their time and consideration. here, we will try to summarize the main concerns outlined by the reviewers and our responses.
>
> **Concern 1 -  What do we mean by adaptive?**
>
> **Response:**
>
> GeLoRA framework recognizes that the parameters adjusted during fine-tuning should not be uniformly distributed across all layers or fixed for all tasks. This is because the model may already possess prior knowledge relevant to the task, either from pretraining data or from earlier fine-tuning on related knowledge. Similarly, each layer contains distinct parameters that capture task-specific features. Hence, by “adaptive” ranks, we mean that the ranks we use for fine-tuning depend on the model, the dataset, and the transformer block, and they are computed prior to fine-tuning. We refer the reviewer to the description of our approach in the revised version, and also the computed rank patterns for the GLUE benchmark in the Appendix.
> However, we do not change the ranks during the fine-tuning process, as we believe that our lower bound will remain optimal. This is because gradient-based techniques like SGD tend to prefer flatter minima, meaning as the training progresses, the number of directions influencing the optimization process, and hence the ranks of the transformer blocks decreases and approach our lower bound. Consequently, the bound we compute becomes increasingly tight and optimal as training advances.
>
> **Concern 2 -  Missing Main Algorithm**
>
> **Response:**
> We acknowledge that the original version of our paper lacked a detailed description of the main GeLoRA framework and its connection to the underlying theory. This has been addressed in the revised version. To summarize, the GeLoRA framework recognizes that the parameters adjusted during fine-tuning should not be uniformly distributed across all layers or fixed for all tasks. This is because the model may already possess prior knowledge relevant to the task, either from pretraining data or from earlier fine-tuning on related knowledge. Similarly, each layer contains distinct parameters that capture task-specific features.
> Building on our theoretical results, we derive a lower bound for the optimal rank of each transformer layer based on the input and output dimensions of the hidden states. This lower bound serves as the rank configuration for fine-tuning.
>
> The algorithm comprises four main steps:
>
> - **Compute hidden states:** Extract the hidden states of the training data from the model.
> - **Estimate intrinsic dimensions:** Use the 2-NN algorithm, a widely adopted method for analyzing deep learning representations, to estimate the intrinsic dimensions of the hidden states.
> - **Determine rank configuration:** Set the rank of each layer based on the computed intrinsic dimensions and our main Theorem. To ensure no significant information is overlooked, an offset of 1 is added to each rank, as the estimated intrinsic dimensions are lower bounds of the true intrinsic dimension.
> - **Fine-tune with LoRA:** Fine-tune the model using LoRA, using the precomputed rank pattern.
>
> This approach ensures a task-specific and layer-adaptive fine-tuning process, aligning with the theoretical analysis of GeLoRA.
>
> **Concern 3 -Experimental Part**
>
> **Response:**
> - We acknowledge that our experiments were conducted solely on the DeBERTaV3 architecture. While this model was introduced in 2021, it remains highly competitive and relevant, particularly for natural language understanding (NLU) tasks like those in the GLUE benchmark. **DeBERTaV3** was specifically designed to excel in these tasks by using techniques such as disentangled attention providing a good foundation for evaluating our framework, especially that other works (LoRA, SoRA, AdaLoRA) have used the same model for their evaluations.
> - Similarly to these works, we conducted additional experiments on question-answering tasks using the **SQuAD dataset (v1.1 and v2.0)** and have included the results in the updated version of our paper.
> - **QQP** and **MNLI** datasets are being added to the revised version.
> - Furthermore, we are actively working on expanding our evaluation by incorporating an additional experiment focused on instruction-tuning tasks and decoder-only models. Specifically, we are fine-tuning the **Phi-2 model** using the **Airoboros** dataset and evaluating its performance on the **MT-Bench** dataset.
> - We have expanded our comparison by including additional baselines, specifically adapter-based techniques such as **Houlsby** and **Pfeiffer** adapters, as well as **AdaLoRA**. However, for **SaLoRA** and **ALoRA**, we were unable to include them as baselines due to the unavailability of publicly accessible code to reproduce their experiments.

---

> ### Author Response · Authors · 2024-11-27
> **Summary of rebuttal revision (2/2)**
>
> **Concern 4 -How do we compute the intrinsic dimensions?**
> **Response:**
> - The computation of the intrinsic dimension of the hidden states is a pre-processing step and, therefore, does not affect the training time, as the ranks are precomputed prior to training.
> - Our updated paper version gives an overview of the 2-NN estimator, and outlines the main GeLoRA algorithm.
> - The complexity of the 2-NN algorithm for estimating intrinsic dimensions is O(LN^2), where L is the number of transformer blocks and N is the number of data samples. However, this complexity can be significantly reduced to linear O(LN) by employing persistent homology-based dimension estimation accelerated using GPUs. This ensures that the pre-processing step remains scalable, even for large datasets and models.
> - The 2-NN algorithm has already been employed in the context of NLP applications to analyze text data and internal transformer representations. This method provides a simple yet effective approach to estimating the intrinsic dimension of hidden representations. We refer the reviewers to the following references, which highlight the use of this technique in NLP and transformer architectures.
>      - Lucrezia Valeriani, Diego Doimo, Francesca Cuturello, Alessandro Laio, Alessio Ansuini, & Alberto Cazzaniga. (2023). The geometry of hidden representations of large transformer models.
>     - Fan Yin, Jayanth Srinivasa, & Kai-Wei Chang. (2024). Characterizing Truthfulness in Large Language Model Generations with Local Intrinsic Dimension.
>    - Eduard Tulchinskii, Kristian Kuznetsov, Laida Kushnareva, Daniil Cherniavskii, Serguei Barannikov, Irina Piontkovskaya, Sergey Nikolenko, & Evgeny Burnaev. (2023). Intrinsic Dimension Estimation for Robust Detection of AI-Generated Texts.
>   - Facco, E., d’Errico, M., Rodriguez, A., & Laio, A. (2017). Estimating the intrinsic dimension of datasets by a minimal neighborhood information. Scientific Reports, 7(1).
> - Additionally, the algorithm can be applied to subsets of the data. In fact, it is often advisable not to use the entire dataset to avoid noisy estimations, as explained by the authors of the original paper.
>
> **Concern 5 - Confusions regarding the mean rank**
>
> **Response:**
> The mean ranks reported in Table 2 were actually the rounded mean ranks which we determined in order to set the rank values for LoRA and its variants and ensure a fair comparison.
>
> **Concern 6 - Mathematical Notations**
>
> **Response:**
> 1.  The dimension of the image of $f$, denoted as $\dim Imf$, represents the rank of the linear transformation $f$, which is the dimension of its image. We used this example to illustrate the intuition behind GeLoRA. Specifically, the dimension of the image of the input under the transformation $f$ is equal to the rank of the matrix $A$. This establishes a connection between the parameters, represented by the matrix $A$, and the data space, represented by $X$, and its transformation under the application of $f$, denoted as $f(X)$. This relationship highlights how the structure of the parameters influences the data space
> 2. 3. We fully agree with the definition provided. However, the expression we presented is in matrix form, where $\phi$ represents the vector of all model parameters. We acknowledge that it is also possible to represent this by specifying the actual values of the matrix entries, as suggested by the reviewer. For reference, we adopted a similar notation to that used by **Shunshi Amari** in his book **Information Geometry and its Applications**.
> 4. In our previous version of the manuscript, we provided the following definition of intrinsic dimension: "The intrinsic dimension is defined as the minimum number of parameters required to capture the local variance of the data points effectively." Within the context of the parameter manifold, the intrinsic dimension is determined by thresholding the eigenvalues of the Fisher Information Matrix (FIM) to exclude uninformative directions. The number of significant eigenvalues, which correspond to informative directions, defines the intrinsic dimension of the parameter manifold. Furthermore, in the revised version of the manuscript, we have clarified our approach to estimating the intrinsic dimension of hidden states using the Two Nearest Neighbors (TwoNN) technique.
> 5. We are adding mathematical formalism section in our Appendix to outline the main definitions, theorems, and algorithms used in our paper.

---

> ### Author Response · Authors · 2024-11-27
> **GeLoRA - Main objective of our work**
>
> Although large language models excel at practical tasks, they lack a strong theoretical foundation. Their abilities to generalize, represent knowledge, and perform complex reasoning remain poorly understood due to opaque training dynamics in high-dimensional parameter spaces. This opacity makes it challenging to explain their behavior, ensure robustness, and systematically improve their design. In our GeLoRA work, we aim to bridge this gap by providing a solid theoretical framework that elucidates the inner working mechanism of LLMs and fine-tuning techniques. We believe that understanding the relationship between the data manifold and training dynamics in parameter space is essential for effective algorithm optimization.
>
> While GeLoRA may slightly underperform on specific tasks, it consistently delivers strong results across diverse benchmarks, achieving higher average scores than other fine-tuning methods. Moreover, it accomplishes this while utilizing significantly fewer parameters, making it a more efficient and scalable solution.

---

### Meta-Review · Area_Chair_aLqE · 2024-12-15

**Metareview:**

In this paper, the authors studied the connection between the intrinsic dimensionality of hidden representation and the rank of parameters, which is promisingly interesting and important. However, after the rebuttal, most reviewers still have concerns about the gap between the theoretical analysis and the proposed method. The empirical and theoretical results are inconsistent, and thus not convincing based on the current shape of the paper. Therefore, this paper is not ready for publication. The authors are encouraged to revise the paper based on the comments given by the reviewers to make the proposed research stronger for future submission.

**Additional Comments On Reviewer Discussion:**

Reviewers still have concerns about the gap between the theoretical analysis and the proposed method after the rebutal.

---

### Decision · Program_Chairs · 2025-01-22

Reject